# TGF-β induces miR-100 and miR-125b but blocks let-7a through LIN28B controlling PDAC progression

Silvia Ottaviani [1], Justin Stebbing[1], Adam E. Frampton [2,3], Sladjana Zagorac[1], Jonathan Krell[3], Alexander de Giorgio[1], Sara M. Trabulo[4,5], Van T.M. Nguyen[1], Luca Magnani [1], Hugang Feng[6], Elisa Giovannetti[7,8], Niccola Funel[8], Thomas M. Gress[9], Long R. Jiao[2], Ylenia Lombardo[1], Nicholas R. Lemoine[10], Christopher Heeschen[4,5] & Leandro Castellano [1,11]

TGF-β/Activin induces epithelial-to-mesenchymal transition and stemness in pancreatic ductal adenocarcinoma (PDAC). However, the microRNAs (miRNAs) regulated during this response have remained yet undetermined. Here, we show that TGF-β transcriptionally induces MIR100HG lncRNA, containing miR-100, miR-125b and let-7a in its intron, via SMAD2/3. Interestingly, we find that although the pro-tumourigenic miR-100 and miR-125b accordingly increase, the amount of anti-tumourigenic let-7a is unchanged, as TGF-β also induces LIN28B inhibiting its maturation. Notably, we demonstrate that inactivation of miR-125b or miR-100 affects the TGF-β-mediated response indicating that these miRNAs are important TGF-β effectors. We integrate AGO2-RIP-seq with RNA-seq to identify the global regulation exerted by these miRNAs in PDAC cells. Transcripts targeted by miR-125b and miR-100 significantly overlap and mainly inhibit p53 and cell–cell junctions' pathways. Together, we uncover that TGF-β induces an lncRNA, whose encoded miRNAs, miR-100, let-7a and miR-125b play opposing roles in controlling PDAC tumourigenesis.

[1] Department of Surgery and Cancer, Division of Cancer, Imperial College London, Imperial Centre for Translational and Experimental Medicine (ICTEM), London W12 0NN, UK. [2] Department of Surgery and Cancer, HPB Surgical Unit, Imperial College, Hammersmith Hospital Campus, London W12 0HS, UK. [3] Department of Surgery and Cancer, Division of Cancer, Imperial College London, Institute of Reproductive and Developmental Biology (IRDB), London W12 0NN, UK. [4] Stem Cells & Cancer Group, Spanish National Cancer Research Centre (CNIO), Madrid 28028, Spain. [5] Stem Cells in Cancer & Ageing, Barts Cancer Institute, Queen Mary University of London, London EC1M 6BQ, UK. [6] Epigenetics and Genome Stability Team, The Institute of Cancer Research, 237 Fulham Road, London SW3 6JB, UK. [7] Department of Medical Oncology, VU University Medical Center, Cancer Center Amsterdam, Amsterdam 1081 HV, The Netherlands. [8] Cancer Pharmacology Lab, AIRC Start-Up Unit, University of Pisa, Pisa 56126, Italy. [9] Clinic for Gastroenterology, Endocrinology, Metabolism and Infectiology, Philipps-University Marburg, Marburg 35037, Germany. [10] Centre for Molecular Oncology, Barts Cancer Institute, Queen Mary University of London, London EC1M 6BQ, UK. [11] University of Sussex, School of life Sciences, John Maynard Smith Building, Falmer, Brighton BN1 9QG, UK. These authors contributed equally: Silvia Ottaviani, Justin Stebbing. Correspondence and requests for materials should be addressed to L.C. (email: l.castellano@sussex.ac.uk)

Pancreatic ductal adenocarcinoma (PDAC) is a deadly disease with a 5-year survival rate of ~6%[1]. PDAC has a malignant cell population comprising both proliferating and cancer stem cells (CSCs)[2,3]. The majority of tumors (95%) are driven by mutational hyper-activation of *KRAS*. Additional characteristics include inactivation of *TP53* (74%), *P16/INK4A* (35%), *SMAD4* (31%), and other TGF-β effectors[4,5]. TGF-β signaling has a vital role in PDAC and other cancers[6]. It is released from the inflammatory tumor microenvironment, and acts as either a tumor suppressor or an oncogene, depending on cellular context[7,8]. It activates SMAD2/3 transcription factors (TFs), which in turn interact with SMAD4 to regulate the transcription of a subset of genes[9] that can differ depending on an individual cell's characteristics[8]. At some cell stages, TGF-β reduces cell

proliferation and increases apoptosis[6]. This effect can be important for PDAC progression, because inactivation of TGF-β signaling components in pancreatic precursor lesions, combined with *KRAS* hyper-activation, induces PDAC formation and metastasis[10,11]. In contrast, TGF-β family members can also promote epithelial-to-mesenchymal transition (EMT), tumourigenesis and metastasis at more advanced stages of the disease[12,13].

We and others have demonstrated that miRNA dysregulation plays a significant role in PDAC tumourigenesis and progression[14–19]. Notably, miRNAs can be important in PDAC stemness and EMT because ZEB1, a transcriptional repressor of CDH1, also inhibits miR-200 family members, as well as miR-203, which in turn repress several inducers of tumourigenesis[17]. Similar to

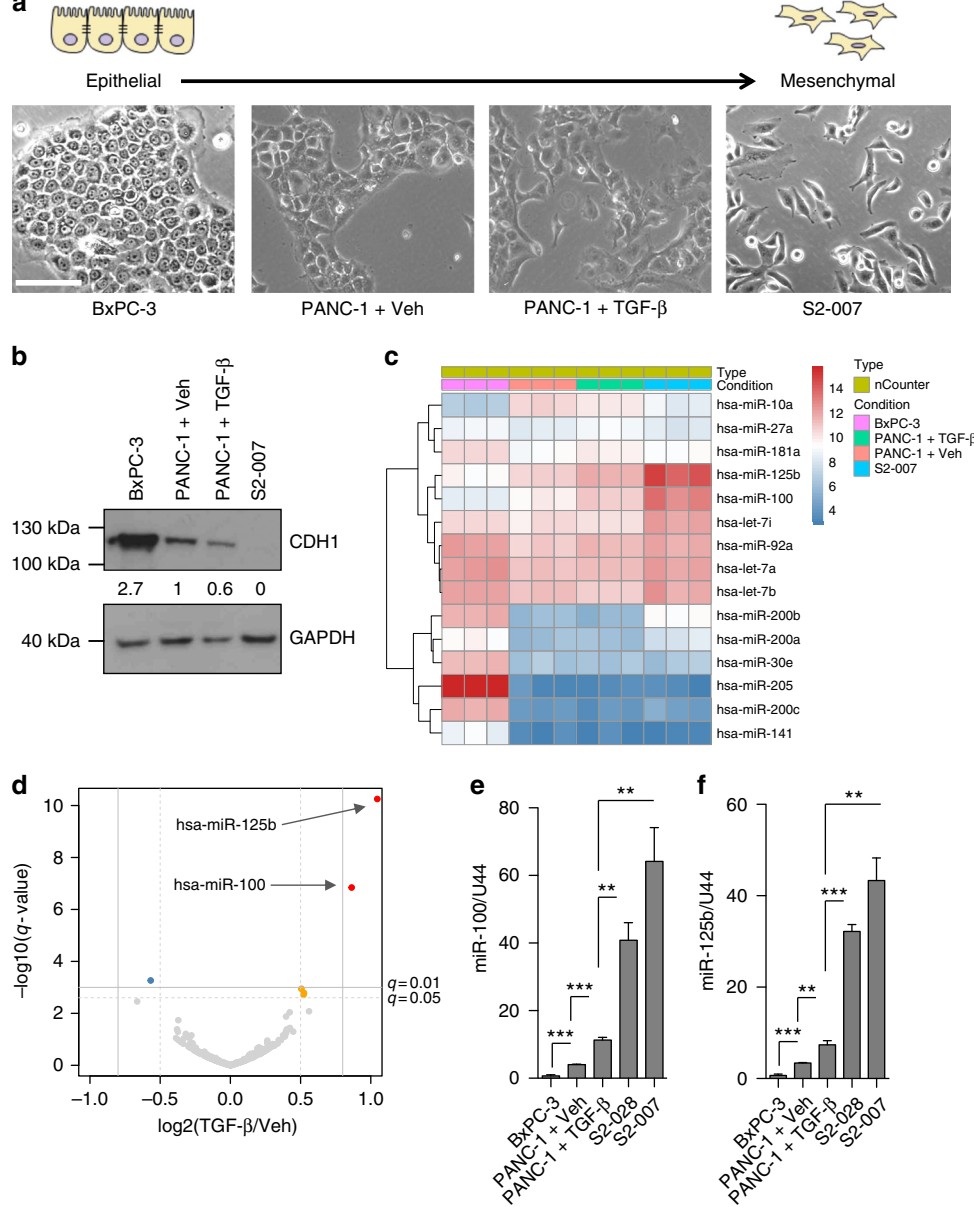

**Fig. 1** miR-100 and miR-125b expression is associated with EMT in PDAC cells. **a** In vitro cell line model of EMT spectrum. Phase-contrast images show PDAC lines ordered from the most epithelial-like (left) to the most mesenchymal-like (right). Scale bar: 100 μm. **b** Immunoblot of E-cadherin (CDH1) in PDAC cell lines with quantification of CDH1/GAPDH ratio. **c** Heatmap showing selected up and down-regulated miRNAs from the nCounter expression analysis. **d** Volcano plot displaying miR-100 and miR-125b being the most significantly up-regulated miRNAs upon TGF-β treatment. **e**, **f** RT-qPCR for miR-100 in **e** and miR-125b in **f** in PDAC lines. Values were normalized to RNU44 levels and are shown as mean ± s.e.m. Results are from three independent experiments each performed in triplicate. **P-value < 0.01, ***P-value < 0.001. P-values were calculated using two-tailed Student's t test

miR-200s, the let-7 family of miRNAs induces reversion of EMT in Gemcitabine (GEM)-resistant PDAC cells[20]. Interestingly, LIN28B has been shown to inhibit the biogenesis of let-7 family members, enhancing the progression of PDAC and other cancers[21,22]. In contrast to miR-200 and let-7 family members, miR-100 and miR-125b are up-regulated in GEM-resistant cells and promote EMT in PDAC[23–25].

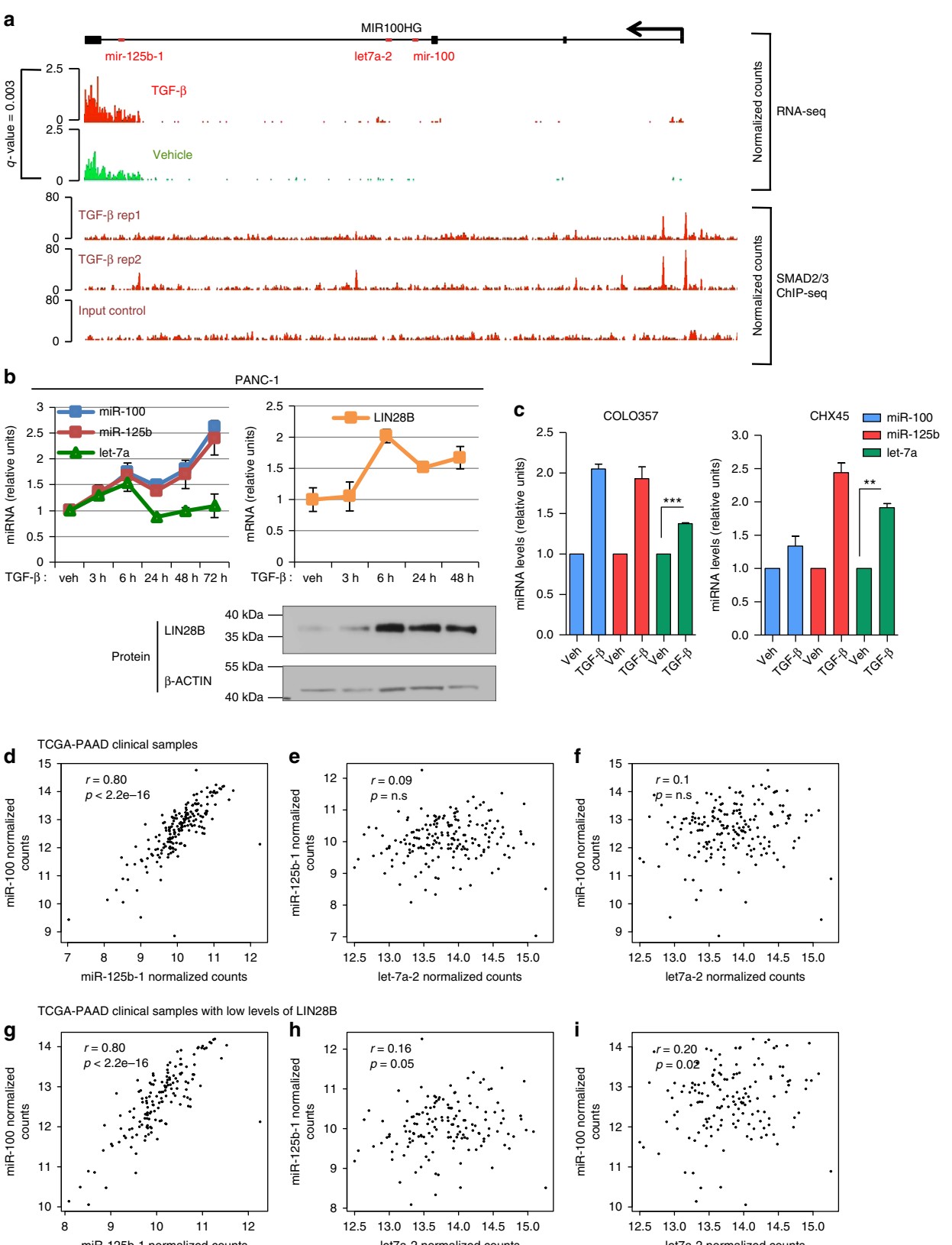

Remarkably, the miRNAs regulated by TGF-β in PDAC have remained undetermined.

Here, we show that TGF-β increases MIR100HG transcription through SMAD2/3. The induction of LIN28B in the same TGF-β response results in the up-regulation of miR-100 and miR-125b, with let-7a unchanged despite being part of the same MIR100HG primary transcript. We also show that these miRNAs regulate a multitude of genes involved in the inhibition of p53 and DNA damage response pathways, which are crucial for the progression of this frequently metastatic disease. Considering that targeting miRNAs could be used for anti-cancer therapy (reviewed in ref. [26]), the inhibition of miR-125b and/or miR-100 in patients could be considered as a new therapeutic approach for treating PDAC, and also as biomarkers for stratifying PDAC.

## Results

**TGF-β treatment induces miR-100 and miR-125b.** To discover novel miRNAs implicated in PDAC progression through TGF-β, we created an in vitro cellular model with cell lines positioned along a gradient moving from epithelial-like to mesenchymal-like status, including cells treated with TGF-β (Fig. 1a), and performed nCounter miRNA expression profiling (Supplementary Data 1). Specifically, we used epithelial-like BxPC-3 cells; PANC-1 cells that are part-epithelial, part-mesenchymal-like; PANC-1 treated with TGF-β that adopt a more spindle-shaped, mesenchymal-like morphology and finally highly invasive/metastatic S2-007 PDAC cells (Fig. 1a). As expected, the expression levels of CDH1 were inversely correlated with the mesenchymal-like status of the cells (Fig. 1b). Additionally, we confirmed that miR-200 family members were strongly down-regulated in mesenchymal-like cells compared to BxPC-3 epithelial-like cells (Fig. 1c and Supplementary Data 1), as previously shown[17,20]. Surprisingly, the expression of this family of miRNAs did not change upon TGF-β treatment in PANC-1 (Fig. 1c and Supplementary Data 1), indicating that they are not part of the TGF-β regulated EMT response in PDAC. Only two miRNAs, namely miR-100 and miR-125b, increased proportionally with the mesenchymal status of the cell (Fig. 1c and Supplementary Data 1), and were significantly up-regulated by TGF-β (adjusted $P < 0.01$, Wald Test) (Fig. 1c, d and Supplementary Data 1). We validated this result by RT-qPCR (Fig. 1e, f). Moreover, the expression of both miR-100 and miR-125b was significantly higher in PANC-1 stably overexpressing TGF-β[27] compared to PANC-1 stably transfected with empty vector, whilst the levels of miR-200 remained unchanged (Supplementary Fig. 1a), independently confirming our findings.

**TGF-β increases MIR100HG transcription through SMAD2/3.** Next, we treated PANC-1 cells with TGF-β and performed RNA-sequencing (RNA-seq) to evaluate how the TGF-β-mediated increase of miR-100 and miR-125b relates to the mRNA regulation exerted by the same stimulus (Supplementary Fig. 1b and Supplementary Data 2). As expected, TGF-β significantly up-regulated pro-EMT factors, such as SNAI1, SNAI2, and CDH2, and down-regulated the cell–cell junction protein CDH1

(adjusted $P < 0.05$, Wald Test, Supplementary Fig. 1c and Supplementary Data 2), supporting the validity of our approach. Interestingly, MIR100HG, a long noncoding RNA (lncRNA) and also the tricistronic host gene of miR-100, miR-125b, and let-7a (Supplementary Fig. 2a), was amongst the RNAs significantly up-regulated by TGF-β treatment (adjusted $P < 0.01$, Wald Test) (Fig. 2a and Supplementary Data 2), indicating transcriptional induction of miR-100 and miR-125b by TGF-β. Accordingly, miR-100 and miR-125b precursors (pre-miR-100 and pre-miR-125b) show a similar pattern of regulation as their mature miRNA forms (Supplementary Fig. 2b). Subsequently, we performed SMAD2/3 chromatin immunoprecipitation sequencing (ChIP-seq) following TGF-β treatment in PANC-1 cells. Regions within 20 kilobases from the transcription start sites (TSSs) of TGF-β up-regulated genes were enriched in SMAD2/3 interaction sites (2.9 fold; $P < 0.0001$, Fisher's exact test) (Supplementary Fig. 2c), indicating that TGF-β regulates several genes through its canonical pathway during EMT in PANC-1 cells. ChIP-seq of SMAD2/3 in both human (Fig. 2a and Supplementary Fig. 3b) and mouse PDAC cells[7] (Supplementary Fig. 2d) indicated that these TFs interact with several regions located along the MIR100HG gene, and have the strongest interactions in proximity to the TSS of the annotated (RefSeq) MIR100HG transcript (Fig. 2a and Supplementary Fig. 3b). Accordingly, the down-regulation of SMAD2 and SMAD3 using specific siRNAs (Supplementary Fig. 2e) significantly reduced miR-100 and miR-125b levels, and totally impaired the ability of TGF-β to increase their abundance (Supplementary Fig. 2f). In addition, MIR100HG represents the only regulated transcript by TGF-β located within its related topological associated domain (TAD) (Supplementary Fig. 3a), indicating that SMAD2/3 interact here to specifically increase MIR100HG transcription. Interestingly in mice PDAC cells, SMAD2/3 interaction was reduced in SMAD4 (−) cells (Supplementary Fig. 2d). Accordingly, SMAD4 has been shown to be important for the regulation of EMT-related genes[7]. We additionally validated the direct SMAD2/3 interaction with the MIR100HG promoter in both COLO357 and PANC-1 PDAC cells by ChIP-qPCR (Supplementary Fig. 2g). Interestingly, MIR100HG transcript appears to have at least three TSSs, as demonstrated by the presence of two additional H3K27ac and H3K4me3 peaks downstream of the annotated TSS (Supplementary Fig. 3b). In order to elucidate transcriptional regulation of MIR100HG by TGF-β, we used CRISPR-Cas9 to remove the first TSS containing the strongest SMAD2/3 interactions (Supplementary Fig. 3b). Although this deletion significantly reduced MIR100HG, as well as miR-100 and miR-125b expression (Supplementary Fig. 3c), it did not impede the ability of TGF-β to increase their levels (Supplementary Fig. 3d), suggesting the intriguing possibility that SMAD2/3 may use other sites more intensively to regulate MIR100HG expression, in the absence of the main ones. In aggregate, these findings suggest that TGF-β activates SMAD2/3, which in turn directly regulates the transcription of a gene network which includes MIR100HG, along with miR-100 and miR-125b, thereby promoting EMT and tumourigenesis in PDAC cells.

**Fig. 2** TGF-β induces miR-100 and miR-125b but not let-7a by co-regulation of MIR100HG and LIN28B. **a** Top: integrative genomics viewer (IGV) tracks show up-regulation of MIR100HG transcript in PANC-1 cells treated with TGF-β for 72 h compared to vehicle control as determined by RNA-seq ($n = 3$ for each condition). Bottom: IGV tracks show SMAD2/3 binding to MIR100HG locus in cells treated with TGF-β for 1 h as determined by ChiP-seq ($n = 2$ for TGF-β treated samples and $n = 1$ for input sample). **b** RT-qPCR of miR-100, miR-125b, let-7a normalized to RNU44 levels (left), LIN28B normalized to GAPDH levels (right) and Immunoblot for LIN28B and β-ACTIN (bottom-right) in PANC-1 cells treated with TGF-β for the indicated time points. **c** RT-qPCR of miR-100, miR-125b and let-7a normalized to RNU44 levels in human PDAC COLO357 and CHX45 cells (derived from a KPC mouse model), after treatment with TGF-β for 72 h. In **b**, **c** values represent the mean ± s.e.m. Results are from three independent experiments each performed in triplicate. *$P$-value < 0.05, **$P$-value < 0.01, ***$P$-value < 0.001. $P$-values were calculated using two-tailed Student's $t$ test. **d**–**i** Pearson correlation analyses in PDAC samples derived from TCGA-PAAD ($n = 183$)

**TGF-β induces miR-100 and miR-125b, but blocks let-7a**. It has been shown that let-7 family members reduce tumourigenesis in PDAC[28] and other cancers[29,30]. Since TGF-β and other family members[13] act to increase pancreatic tumourigenesis, any rise in let-7a levels following MIR100HG induction as part of this response would serve to counteract this effect. In fact, despite the TGF-β-mediated increase in MIR100HG transcription, which augments miR-100 and miR-125b levels (Figs 1c–f and 2a, Supplementary Data 1-2), 72 h of TGF-β treatment did not significantly up-regulate either let-7a or other let-7 family members

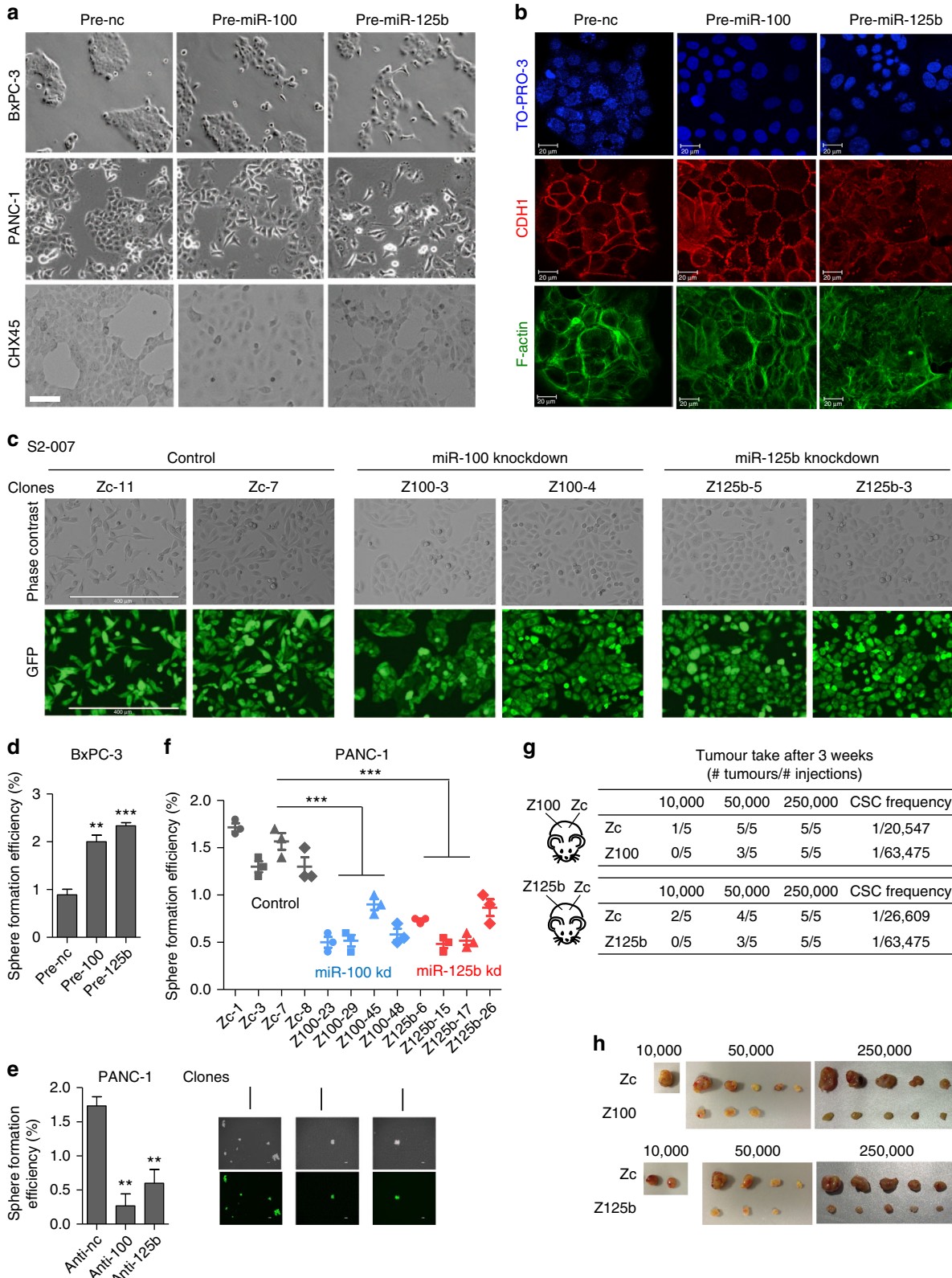

in PANC-1 cells (Supplementary Data 1). Conversely, we noticed a slight TGF-β-mediated decrease of let-7b expression (Supplementary Data 1, adjusted $P = 0.08$, Wald Test). This suggests that let-7a is repressed at the post-transcriptional level by another TGF-β-regulated factor, thereby remaining unchanged during the response. To identify the mechanism of let-7a regulation we interrogated our RNA-seq data to see if factors known to regulate let-7 biogenesis (reviewed in ref.[31]) were modulated by TGF-β (Supplementary Data 2). Strikingly, LIN28B, implicated in the post-transcriptional inhibition of let-7 maturation[21,32], was significantly induced by TGF-β (Supplementary Data 2, adjusted $P < 0.01$, Wald Test). This suggested that while LIN28B and MIR100HG are both induced by TGF-β, with miR-100 and miR-125b levels increasing, let-7a levels remain low due to post-transcriptional repression by LIN28B. In fact, LIN28B up-regulation has been shown to decrease let-7 levels, and increase let-7 targets during PDAC progression[22]. Accordingly, silencing of LIN28B by RNA interference in PANC-1 cells increased let-7a, but not miR-100 and miR-125b expression (Supplementary Fig. 4a). Likewise, we noticed that PANC3.27 cells treated with shRNAs against LIN28B[22] resulted in increased let-7a, but not miR-125b or miR-100 levels. To evaluate the kinetics of LIN28B/let-7a regulation during the TGF-β response, we performed a time course experiment following TGF-β treatment, and measured miR-100, miR-125b, let-7a levels (Fig. 2b left panel), as well as both LIN28B mRNA (Fig. 2b top-right panel) and protein (Fig. 2b bottom-right panel) levels. Strikingly, while 3 h of TGF-β treatment induced both miR-100 and miR-125b expression, which progressively increased from 3 to 72 h (Fig. 2b left panel), let-7a levels significantly rose from 3 to 6 h, but then returned to untreated levels once LIN28B started accumulating at 6 h (Fig. 2b). Conversely, TGF-β significantly increased let-7a along with miR-100 and miR-125b in two additional TGF-β responsive (Supplementary Fig. 4b) PDAC cells, human COLO357 and mouse CHX45 (isolated from KPC mice by Hermann and colleagues[33]) (Fig. 2c), in which LIN28B levels were almost undetectable (Supplementary Fig 4c). Finally, the capacity of TGF-β to increase the levels of let-7a was restored in multiple PANC-1 cellular clones knocked-out (KO) for LIN28B generated by CRISPR-Cas9 (Supplementary Fig. 4d). LIN28A, homologous protein of LIN28B was never expressed or induced by TGF-β in these cells (Supplementary Fig. 4e), excluding its involvement in this process. Notably, miR-100 and miR-125b expression were closely correlated in 183 PDAC samples derived from The Cancer Genome Atlas (TCGA) ($r = 0.8$, $P < 2.2e-16$, Pearson correlation test), indicating that these two miRNAs are usually co-expressed in PDAC (Fig. 2d). Although neither miR-100 nor miR-125b was correlated with let-7a (Fig. 2e, f), the correlation of miR-100 or miR-125b with let-7a increased and became significant when assessed only in PDACs expressing low levels of LIN28B (Fig. 2h, i).

Interestingly, in contrast to SMAD4 (+) cells (PANC-1, COLO357 and CHX45) (Fig. 2b, c), TGF-β was unable to induce either miR-100 or miR-125b levels in SMAD4 (−) cells (BxPC-3 and S2-007) (Supplementary Fig. 4f), indicating that SMAD4 is crucial for MIR100HG induction.

In aggregate, these data indicated that TGF-β-SMADs-signaling induces MIR100HG and the miRNAs contained within it, but later also induces LIN28B to reduce let-7a levels in order to enhance the TGF-β response.

**miR-100 and miR-125b control PDAC progression**. To evaluate whether miR-125b and miR-100 regulate EMT, tumourigenesis and metastasis, we overexpressed both miRNAs in BxPC-3, PANC-1, and mouse CHX45 cells, and assessed both cell morphology (Fig. 3a) and levels of EMT markers (Supplementary Fig. 5a). Overexpression of each miRNA produced more spindle-shaped cells than cells transfected with miRNA negative control (n.c.), a characteristic of mesenchymal cell types (Fig. 3a), thus confirming previous findings[23]. While both miRNAs may stimulate EMT in these cells, the effect of miR-125b seemed more marked (Fig. 3a). Both miRNAs may act by elevating the expression of the mesenchymal marker VIM, but without significantly reducing CDH1 levels (Supplementary Fig. 5a). Despite this, immunofluorescent experiments revealed that cell–cell interaction was impaired. We observed that miR-100 induced disruption of CDH1 at the level of cell–cell junctions (Fig. 3b), while miR-125b promoted a complete CDH1 and actin delocalization from the junctions (Fig. 3b). Next, we stably impaired the activity of miR-100 or miR-125b using miRZip technology in metastatic/mesenchymal-like S2-007 cells expressing high levels of both miRNAs (Supplementary Data 1). In contrast to overexpression experiments (Fig. 3a,b), clones of S2-007 stable cell lines (Z100-3, Z100-4, Z125b-5, Z125b-3) with reduced miR-100 or miR-125b activity adopt remarkable epithelial-like morphologies (Fig. 3c). Additionally, mesenchymal markers such as VIM, ZEB1, and SNAI1 were significantly reduced (Supplementary Fig. 5b). This suggested that inhibition of these miRNAs in mesenchymal-like PDAC cells could promote mesenchymal to epithelial transition (MET), independent of CDH1 levels.

It is known that EMT forms cells that are more motile and metastatic[17]. Wound-healing scratch assays, followed by video tracking of cell movements, showed that S2-007 cells stably knocked down for miR-100 or miR-125b (Supplementary Fig. 5c and Supplementary Movies 1–3), as well as LPC006 and LPC067 primary PDAC cultures transfected with anti-miR-100 and anti-miR-125b (Supplementary Fig. 5d–g), displayed a strong reduction in cell migration compared to controls. Moreover, S2-007 cells with impaired miR-125b activity had markedly reduced colonization of mice livers (i.e. metastatic spread), whilst S2-007

**Fig. 3** Effects of miR-100 and miR-125b on EMT and stemness. **a** Phase-contrast images of BxPC-3, PANC-1, and CHX45 cells treated with control mimic (pre-nc) or mature miRNA mimics (pre-miRs) for 12 days (5 nM). Scale bar: 100 µm. **b** Immunofluorescence (IF) staining for CDH1 (red) and F-actin (green) in BxPC-3 treated as in **a**. Nuclei are visualized with TO-PRO-3 stain (blue); scale bar: 20 µm. In **a**, **b** representative images from three independent experiments are shown. **c** Morphology of S2-007 stable knockdown clones for miR-100 (Z100), miR-125 (Z125b), and control (Zc) generated with Zip technology. Representative pictures from two different clones per treatment are shown. Phase-contrast images are shown at the top and corresponding GFP images at the bottom. Scale bar: 400 µm. **d**–**f** Sphere-forming assays in BxPC-3 cells transiently transfected with precursor miRNA mimics (**d**), in PANC-1 cells treated with anti-miRNA inhibitors (**e**), and in four independent PANC-1 Zip stable knockdown clones ($n = 4$) (**f**). Representative spheres are shown in lower panels in phase-contrast images and corresponding GFP images. Values represent the mean ± s.e.m. Data are from three independent experiments each performed in triplicate. **g** Mice were subcutaneously injected with the indicated number of S2-007 Zip control cells (Zc-11) on the right flank and S2-007 Zip stable miR-100 knockdown (Z100-3) or miR-125b knockdown (Z125b-5) on the left flank ($n = 5$ per group). Tumor take was determined 3 weeks post-injection. Cancer stem cell (CSC) frequencies were calculated using the extreme limiting dilution analysis algorithm (http://bioinf.wehi.edu.au/software/elda/). **h** Images of resected tumors are shown. **P-value < 0.01, ***P-value < 0.001. P-values were calculated using two-tailed Student's $t$ test

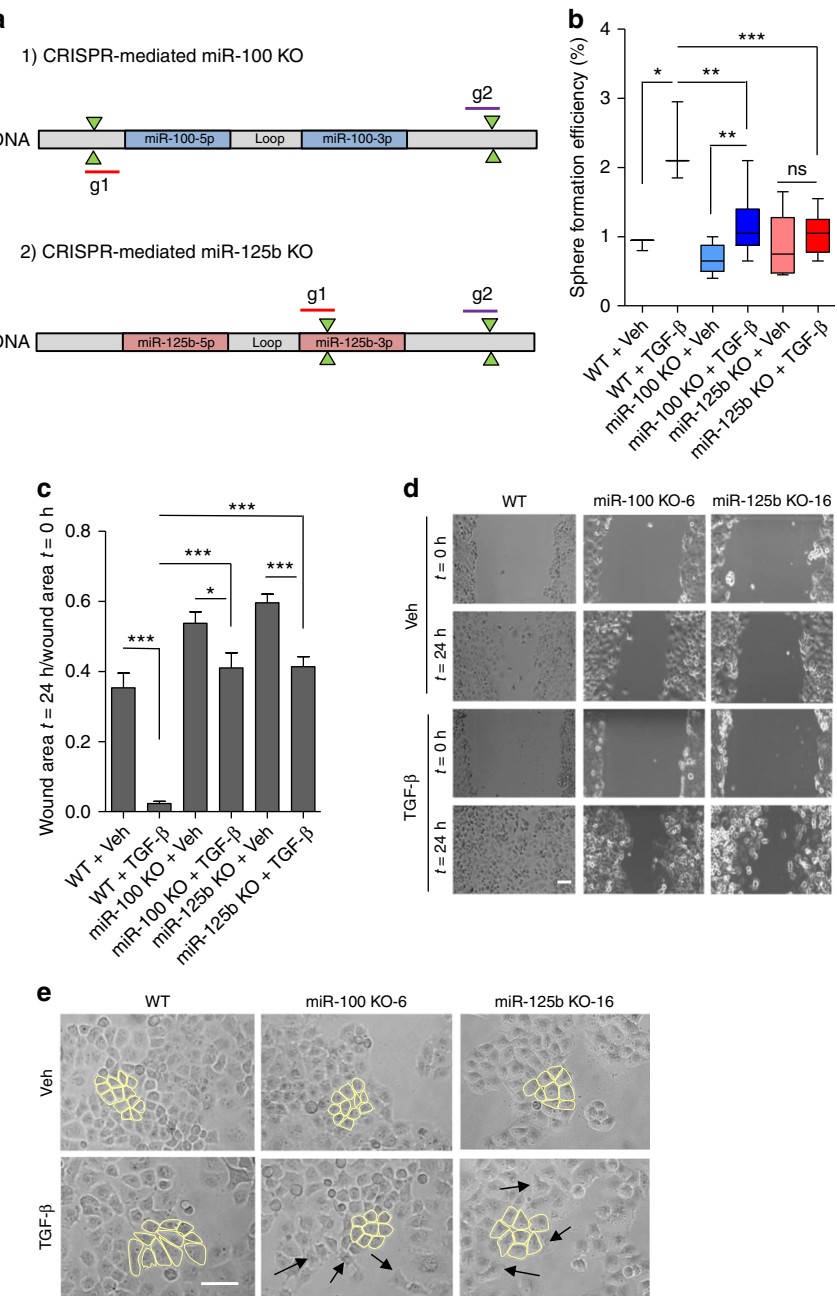

**Fig. 4** miR-100 and miR-125b impairs TGF-β-induced EMT, and stemness. **a** Strategy used to generate CRISPR-Cas9 mediated KO of miR-100 (top) and miR-125b (bottom) in PANC-1 cells. Schematic structure of both miRNA loci are shown. Pairs of sgRNAs were used and are indicated as g1 and g2. **b** Sphere-forming assay in PANC-1 CRISPR-Cas9 KO clones for miR-100 ($n = 3$) and miR-125b ($n = 3$) and in parental wild-type cells (WT). Cells were treated with vehicle (Veh) or TGF-β for 72 h in adherent and then placed in non-adherent conditions for sphere assay. Box plots show median and whiskers are minimum and maximum. Results are from three independent experiments each performed in triplicate. **c** Wound-healing migration assay performed in PANC-1 CRISPR-Cas9 KO clones for miR-100 ($n = 3$) and miR-125b ($n = 3$) and WT cells treated with vehicle (Veh) or TGF-β for 72 h. The wound area at time 0 h and the area left unhealed at 24 h was measure using ImageJ software. The results are presented as a ratio (wound area $t = 24$ h / wound area $t = 0$). Results are shown as mean ± s.e.m. Data are from three independent experiments each performed in triplicate. **d** Representative pictures of the wound-healing assay. Clone 6 for miR-100 KO (KO-6) and clone 16 for miR-125b KO (KO-16) are shown here. Scale bar: 100 μm. **e** PANC-1 WT cells CRISPR-Cas9 KO clones for miR-100 ($n = 3$) and miR-125b ($n = 3$) and WT cells treated with vehicle (veh) or TGF-β for 72 h. Representative phase-contrast images for miR-100 KO-6 and miR-125b-KO16 are shown here. Cell shape of representative cells was manually delineated. Arrows indicate occasional elongated cells in miR-100 and miR-125b KO lines treated with TGF-β. Scale bar: 100 μm. *$P$-value < 0.05, **$P$-value < 0.01, ***$P$-value < 0.001. $P$-values were calculated using two-tailed Student's $t$ test

cells with impaired miR-100 activity were less effective (Supplementary Fig. 6a, b).

Furthermore, it has been demonstrated that EMT can generate cells with properties of stem cells, which are highly tumourigenic and metastatic, as well as resistant to chemotherapy[17,34]. In addition, TGF-β family members induce both EMT and stemness[13,35]. This suggests that TGF-β may increase miR-100 and miR-125b expression to promote both EMT and

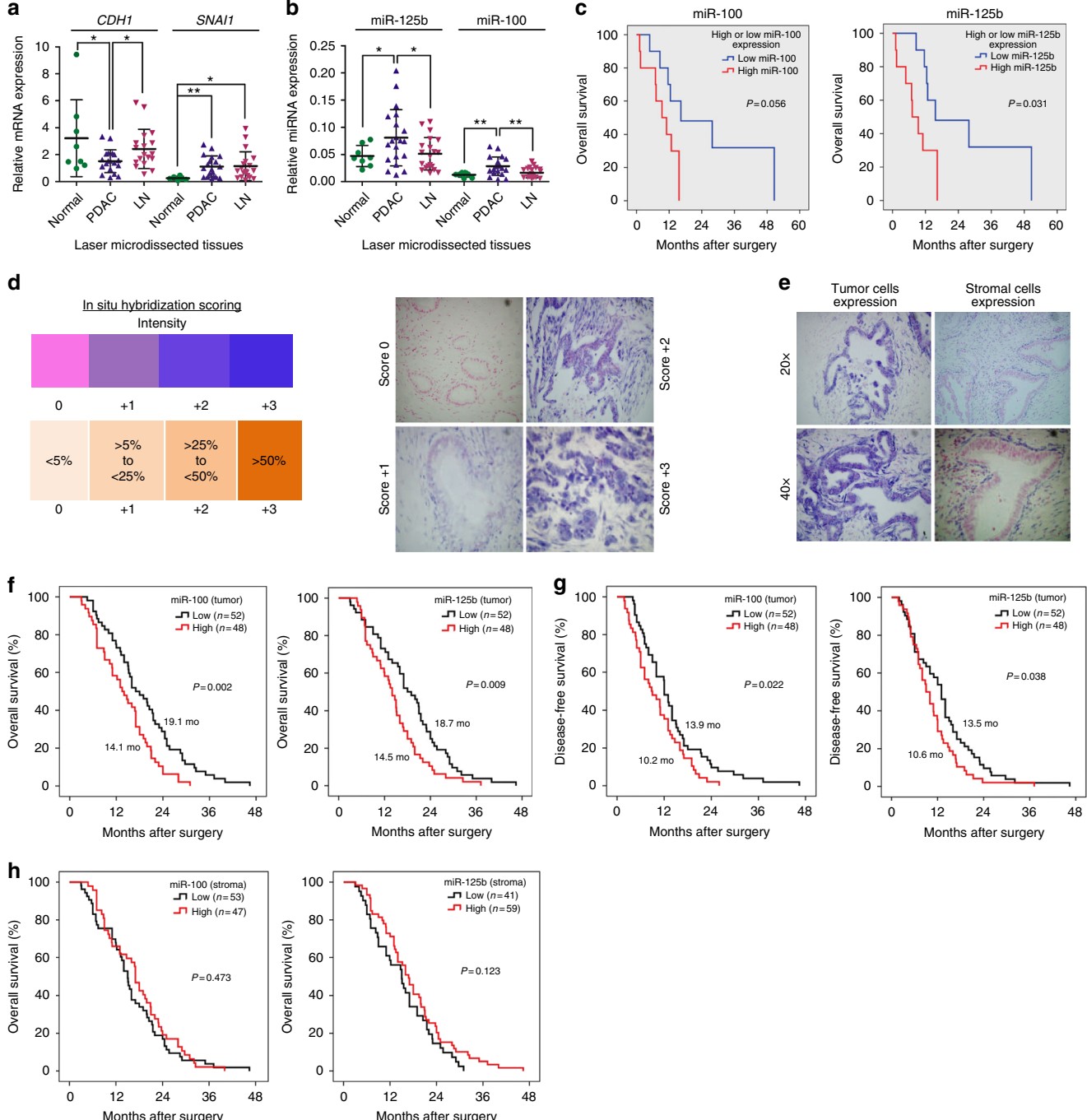

**Fig. 5** miR-100 and miR-125b are up-regulated in PDAC tumors from patients. **a** RT-qPCR for CDH1 or SNAI1 in laser microdissected tissue samples (adjacent normal n = 8, PDAC n = 20, infiltrated lymph-node (LN) n = 20). Results were normalized to GAPDH levels. **b** RT-qPCR for miR-100 or miR-125b in samples described in **a**. Results were normalized to RNU44 levels. In **a**, **b** results are shown as mean of three technical replicates ± s.e.m.* P-value < 0.05, **P-value < 0.01, ***P-value < 0.001. P-values were calculated using two-tailed Student's t test. **c** High levels of miR-100 (left; 9.7 months (95% CI 6.2–13.1) vs. 26.4 months (95% CI 13.5–39.2)) or miR-125b (right; 9.1 months (95% CI 5.4–12.8) vs. 27.1 months (95% CI 14.7–39.4)) are associated with poor overall-survival (OS). **d**, **e** In situ hybridization (ISH) for miR-100 or miR-125b was performed on a tissue-microarray (TMA) of PDAC patients (all Stage IIB; n = 100). ISH scoring system used for miRNA expression analysis (**d**, left). The intensity scale ranges from 0 for no staining to 3 + for the most intense staining. Representative PDAC tumor cores with varying miRNA ISH intensity (**d**, right). Examples of strong miRNA signal in tumor and stromal cells (**e**). **f**, **g** In **f**, left: high miR-100 tumoral expression was associated with reduced OS after surgical resection (14.1 months, 95% CI 12.1–16.1) vs. low miR-100 tumoral expression (19.1 months, 95% CI 16.5–21.7; P = 0.002). Right: high miR-125b tumoral expression was also associated with reduced OS after surgical resection (14.5 months, 95% CI 12.4–16.6) vs. low miR-125b tumoral expression (18.7 months, 95% CI 16.1–21.3; P = 0.009). In **g**, left: high miR-100 tumoral expression was associated with reduced disease-free survival (DFS) after surgical resection (10.2 months, 95% CI 8.4–12.0) vs. low miR-100 tumoral expression (13.9 months, 11.5–16.3; P = 0.022). Right: high miR-125b tumoral expression was associated with reduced DFS after surgical resection (10.6 months, 95% CI 8.8–12.5) vs. low miR-125b tumoral expression (13.5 months, 11.1–15.9; P = 0.038). **h** Expression of miR-100 and miR-125b were not associated with reduced OS after surgical resection. In **f**–**h**, P-values were obtained using log-rank statistics

tumourigenesis in PDAC. To test this hypothesis, we performed tumor-sphere formation assays[13,36] after manipulation of miR-100 or miR-125b activity. Overexpression of miR-100 or miR-125b in BxPC-3 cells, which have low levels of these two miRNAs (Supplementary Data 1), significantly increased the number of tumor spheres (Fig. 3d). Conversely, inhibition of miR-100 or miR-125b with anti-miRNAs in PANC-1 cells, reduced tumor spheres compared to anti-miRNA-n.c. transfected cells (Fig. 3e). Moreover, independent PANC-1 clones with stably impaired activity of miR-100 or miR-125b by miRZip had reduced capacity to form tumor spheres, compared to clones infected with vector controls (Fig. 3f). We were unable to develop individually stable

tumor spheres in culture using S2-007 cells, supporting the notion that highly mesenchymal cells deficient in cell–cell adhesion may fail to form reliable tumor spheres[37]. As an alternative, we assessed expression levels of CD133, an established and specific marker for evaluation of pancreatic CSC enrichment[13,14]. We demonstrated that S2-007 clones with impaired activity of miR-100 or miR-125b expressed significantly lower levels of CD133 (Supplementary Fig. 6c, d). Pancreatic CSCs are notoriously resistant to GEM[13], indicating that treatments aiming to remove this population in conjunction with GEM have the potential to be an effective adjuvant PDAC therapy[13].

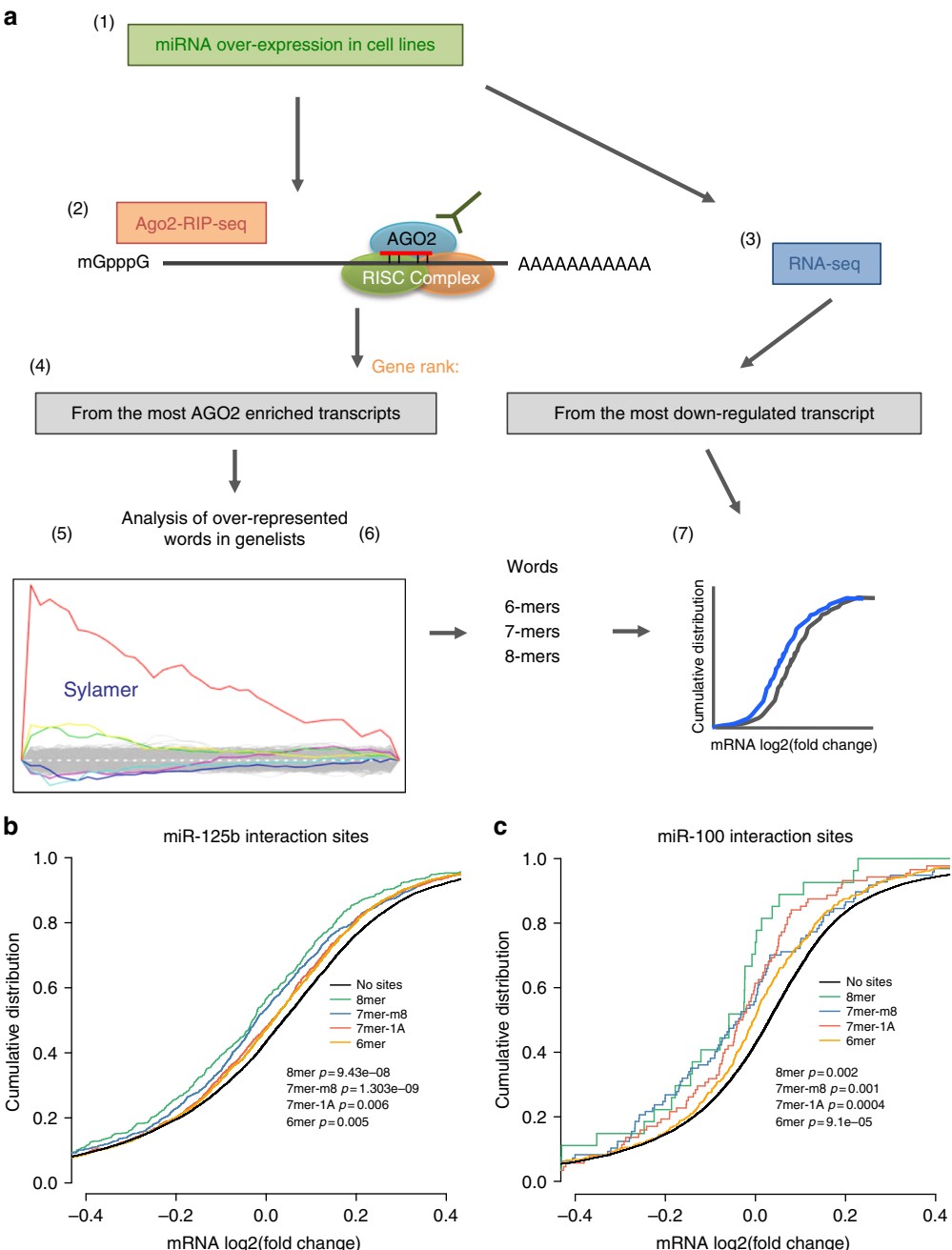

**Fig. 6** RIP-USE identifies miR-125b and miR-100 targets in PDAC. **a** Schematic overview of the steps of the experimental and computational analysis pipeline of RIP-USE for the identification of miRNA targets. **b** The fold change of transcript levels mediated by the overexpression of miR-125b is analyzed comparing transcripts containing 6–8mers with transcripts lacking these motifs in 3′UTRs. **c** The fold change of transcript levels mediated by the overexpression of miR-100 is analyzed comparing transcripts containing 6–8mers with transcripts lacking these motifs in 3′UTRs. *P*-values were calculated by two-sided Kolmogorov–Smirnov test

We confirmed that, similarly to established PDAC cell lines[23,25], anti-miR-100 and anti-miR-125b significantly sensitized two different primary PDAC cell cultures, derived from laser-microdissected PDAC specimens (LPC006 and LPC067), to apoptosis upon GEM treatment (Supplementary Fig. 6e), indicating that these miRNA inhibitors could be clinically useful.

To assess whether miR-100 and miR-125b regulate tumor-initiating capacity of PDAC cells, we performed serial dilution assays in nude mice with S2-007 cells stably knocked down for miR-100, miR-125b, or empty vector control (Fig. 3g, h). Accordingly, these knockdown clones showed a strong reduction in their in vivo tumor-initiating capacity (Fig. 3g, h). Moreover, even when these knockdown cells formed tumor xenografts, they were significantly smaller than empty vector controls (Fig. 3h).

**miR-100 and miR-125b are required for TGF-β cell responses.** We next assessed whether miR-100 and miR-125b were critical for TGF-β-induced tumourigenesis. Remarkably, TGF-β-mediated increase in tumor-sphere formation was reverted in vitro, in independent PANC-1 KO clones for miR-125b, generated by CRISPR-Cas9 (Fig. 4a, b and Supplementary Fig. 7b, d), but was only partially reduced in clones KO for miR-100 (Fig. 4a, b and Supplementary Fig. 7a, c). We observed a similar effect performing in vivo tumourigenic assays using PANC-1 clones with reduced miR-100 and miR-125b activity (Zip clones), with or without TGF-β stable overexpression (Supplementary Fig. 8a). This indicated that miR-125b is essential for TGF-β induced tumourigenesis in PDAC, and represents the most important TGF-β effector amongst the miRNAs derived from MIR100HG.

Importantly, the ability of TGF-β to increase motility (Fig. 4c, d) or formation of spindle-shaped cells (Fig. 4e) was strongly inhibited in the PANC-1 CRISPR-Cas9 clones that were KO for miR-100 and miR-125b. In addition, transient inhibition of both miR-100 and miR-125b negatively affected the capacity of TGF-β to induce the formation of spindle-shaped cells (Supplementary Fig. 8b), or to increase cell motility (Supplementary Fig. 8c) in PANC-1 cells stably overexpressing TGF-β or empty vector control.

**Clinical relevance of miR-100 and miR-125b.** Next, we assessed miR-100 and miR-125b expression in PDAC tissues, by combining data from multiple studies (Supplementary Table 1). This analysis showed that both miRNAs are significantly up-regulated in independent cohorts of PDAC samples versus normal pancreas (NP; miR-125b, PDAC = 343, Normal = 173; miR-100, PDAC = 258, Normal = 129) (Supplementary Table 1). We also used our collection of laser-microdissected samples and measured CDH1, SNAI1 as well as miR-125b and miR-100 using RT-qPCR (Fig. 5a–c). This revealed that CDH1 expression is decreased in PDAC compared to NP samples, but in lymph-node (LN) metastases increase to the levels present in NP, suggesting that LN metastatic cells may have undergone MET (Fig. 5a). In contrast, both miR-100 and miR-125b were significantly up-regulated in PDAC, but in LN metastases decrease to the levels present in NP (Fig. 5b). This suggests that their expression is down-regulated as part of a shift towards a more epithelial-like phenotype as cells establish within the LN. Interestingly, high expression of miR-100 and miR-125b was associated with worse overall-survival (OS) after surgical resection (Fig. 5c). Next, we evaluated whether OS was only associated with miRNA expression in tumor cells, or whether expression in surrounding stroma could also be important. To investigate this, we performed in situ hybridization (ISH) for miR-100 and miR-125b on an independent collection of PDAC samples (n = 100; all Stage IIB) and quantified miRNA expression intensity (Fig. 5d, e). Whilst high miR-100 and miR-

125b levels in tumor cells were associated with both reduced OS (Fig. 5f) and reduced disease-free survival (DFS) (Fig. 5g) after resection, there was no significant association between stromal expression and prognosis (Fig. 5h). Moreover, we found that poorly differentiated (high grade) tumors were associated with high miR-125b (P = 0.042), but not high miR-100 tumoural expression (Supplementary Table 2).

**RIP-USE globally identifies miR-100 and miR-125b targets.** Our results indicated that these two TGF-β-regulated miRNAs are involved in several overlapping phenotypes that could be explained by the regulation of multiple targets. To identify the targets post-transcriptionally regulated by them in PDAC, we developed a novel approach to integrate miRNA overexpression with AGO2 RNA immunoprecipitation (RIP) sequencing (RIP-seq) and differential expression analysis within a bioinformatics framework with Sylamer and cWords algorithms[38–40]. We called this method RIP followed by Unbiased Sequence Enrichment analysis (RIP-USE) (Fig. 6a and Methods). This method is developed in several steps (Fig. 6), including overexpression or down-regulation of the miRNA of interest in cell lines, followed by AGO2-RIP-seq and RNA-seq of total RNA to reveal both transcripts that are significantly enriched (in the case of miRNA overexpression), or depleted (in the case of miRNA inhibition) from AGO2, and are functionally repressed by the miRNA–AGO2–target interaction. This is followed by unbiased seed enrichment analysis to identify ribonucleotide regions of miRNA–transcript interaction (Fig. 6). Since miR-100 and miR-125b were both up-regulated about 40-fold in our highly mesenchymal-like S2-007 cells, compared to the most epithelial-like and less tumourigenic BxPC-3 cells (Fig. 1e, f and Supplementary Data 1), we took this degree of up-regulation to represent a physiologically appropriate range for bridging the EMT spectrum, such that 40-fold overexpression in the TGF-β PANC-1 cells could enable us to identify relevant targets of a miRNA-induced EMT. To this end, we chose the concentration of mimics that increased the cellular miRNA levels by about 40-fold (Supplementary Fig. 9a, b) and performed RIP-USE (Fig. 6a). As expected, 3′UTRs of transcripts that were loaded onto AGO2 after miR-125b or miR-100 overexpression (Supplementary Data 3) were also strongly enriched with miR-100 or miR-125b "seed" motifs (Supplementary Fig. 9c–h). Consistently, cWords showed similar results to Sylamer (Supplementary Fig. 9i, j).

Interestingly, words of nucleotides enriched for miR-100 targets also included U rich motifs (URMs) (Supplementary Fig. 9j), indicating that additional RNA-binding proteins may be important during miR-100 regulation, as has been shown for other miRNAs[39]. To test whether the motifs identified as interacting with AGO2-loaded miRNAs (Supplementary Fig. 9c–j) also inhibit the expression of those genes, we performed cumulative distribution analysis using RNA-seq data obtained following miR-100 or miR-125b overexpression. As expected, transcripts containing 8mer, 7mer-m8, 7mer-1A as well as 6mer seeds were significantly down-regulated compared to transcripts lacking these motifs, thus confirming that interaction of AGO2 with these sites in the 3′UTRs down-regulates the targets in our system (Fig. 6b, c). Transcripts containing 8mer and 7mer-m8 motifs were more greatly suppressed than targets with 7mer-1A and 6mer sites (Fig. 6b, c), confirming previous findings[41].

**Seeds for miR-100 have been depleted during evolution.** To evaluate the molecular pathways regulated by the targets of miR-100 and miR-125b in PDAC, we considered the list of genes overlapping between transcripts loaded onto AGO2 and genes

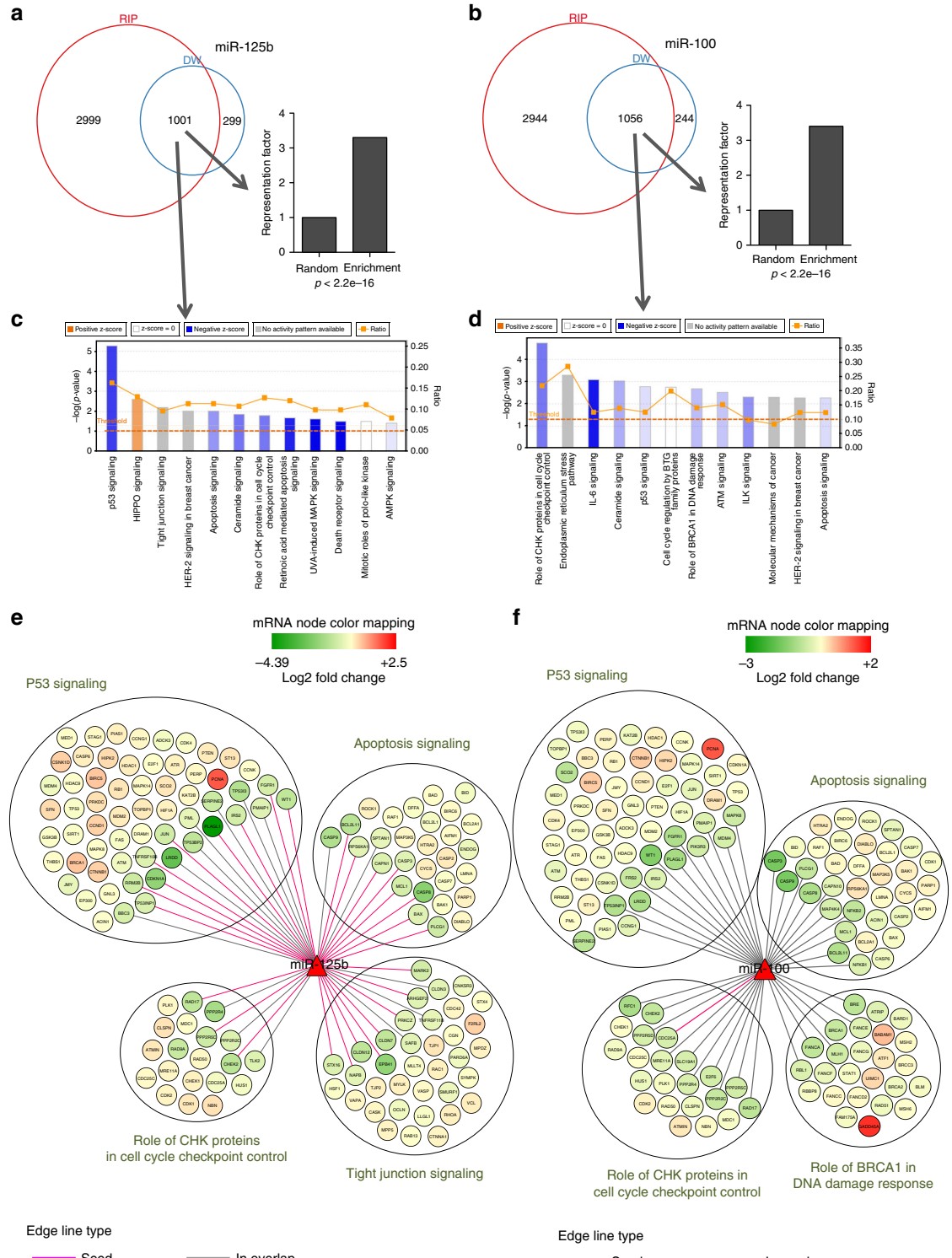

**Fig. 7** miR-100 and miR-125b target similar transcripts to regulate common pathways. **a**, **b** Venn diagrams showing miR-125b (**a**) or miR-100 (**b**) direct targets represented as overlap of the top 4000 transcripts most enriched onto AGO2 from RIP-seq (RIP) and the top 1300 down-regulated transcripts in RNA-seq (DW), after miRNA overexpression. The bar charts indicate the statistical significance of the overlap between the two groups. *P*-values were calculated using exact hypergeometric probability test. **c**, **d** IPA shows the most enriched canonical pathways for miR-125b (**c**) or miR-100 (**d**). Genes from the overlap between RIP and DW were used as input for the analysis. The threshold of significance is indicated by the intermitted line. Positive Z-score (red) indicates that the canonical pathway is activated and negative Z-score (blue) that is inhibited, based on the expression values in the data set. **e**, **f** Sub-networks of genes belonging to significantly enriched IPA canonical pathways using the gene signature coming from overlap between RIP and DW regulated by miR-125b (**e**) or miR-100 (**f**). MiRNA–target interactions inferred from RIP-USE are depicted by different line. Node color represents change in gene expression, from the RNA-seq, mediated by the overexpression of each miRNA

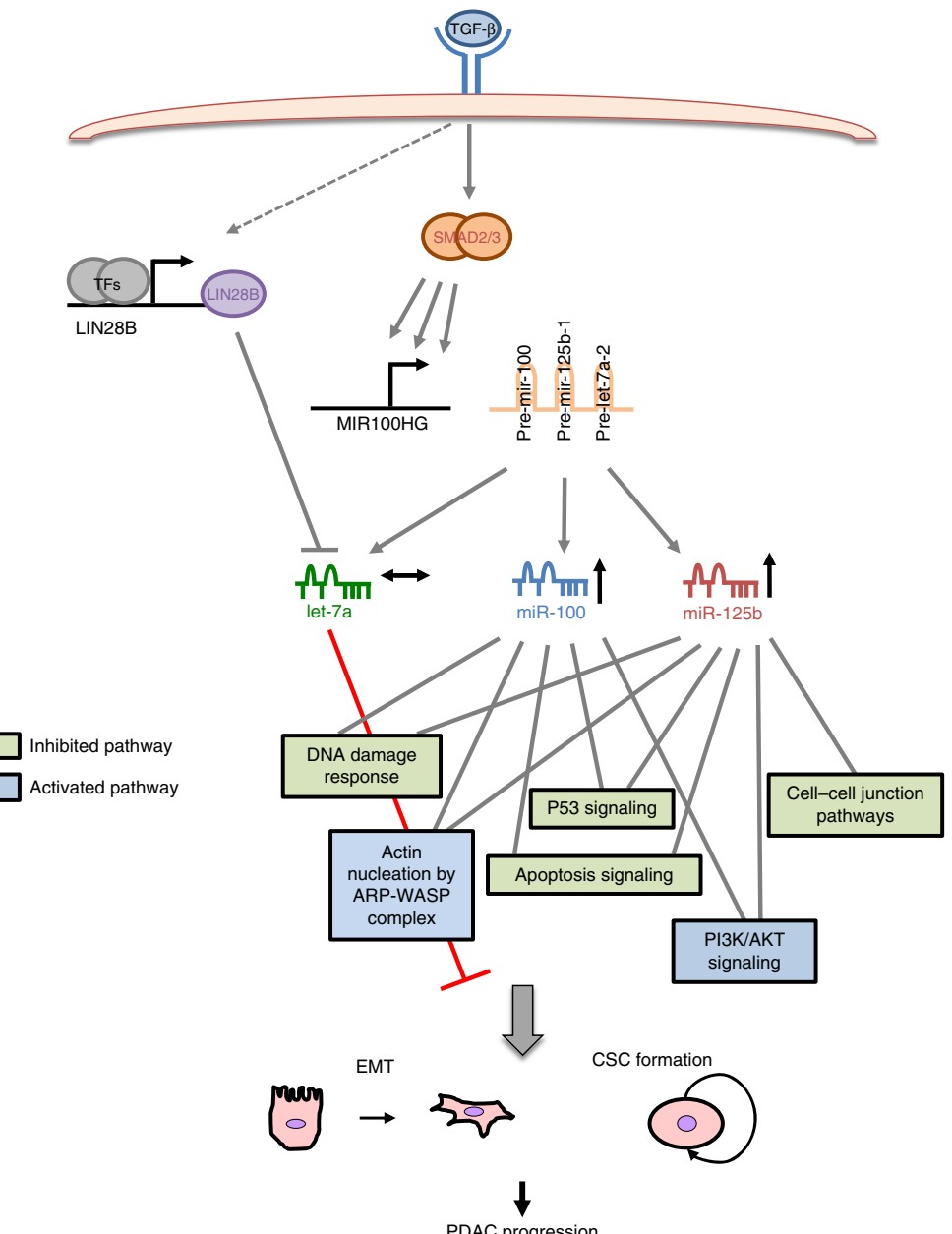

**Fig. 8** Proposed mechanism of action of TGF-β regulated miR-100 and miR-125b in PDAC

down-regulated following miRNA overexpression, to select for directly regulated targets. GSEA analysis of genes down-regulated by miR-125b or miR-100 overexpression (RNA-seq) showed the highest density of 7mer-m8 interacting motifs in the top 1300 down-regulated genes (Supplementary Fig. 10a, b), whereas AGO2-transcripts enrichment analysis upon miR-125b or miR-100 overexpression (RIP-seq) found the highest density of 7mer-m8 interacting sites in the top 4000 most AGO2-loaded transcripts (Supplementary Fig. 10c, d). The overlaps between genes enriched onto AGO2, and genes down-regulated after miRNA overexpression were over 3-fold higher than expected by chance ($P < 2.2e$-16, exact hypergeometric probability) for both miRNAs (Fig. 7a, b), confirming that these fractions are strongly enriched for directly regulated targets. We next validated miRNA-target regulation using both luciferase reporter assays and evaluated their expression levels in PANC-1 clones with impaired activity for each miRNA, selecting important modulators of the

observed phenotypes (Supplementary Fig. 10e–h). MiRNA recognition sites for miR-100 were much fewer than the sites recognized by miR-125b (Supplementary Fig. 10a–d). This was because the number of miR-125b seeds located in all transcripts were higher than the null expectation of randomly generated 7 and 8mers, whereas the number of seeds for miR-100 was considerably lower than that null expectation (Supplementary Fig. 11a). This indicates that although miR-125b seeds in transcripts have been gained during evolution, miR-100 consensus sites have been depleted. Despite this remarkable difference, the number of genes that overlap between RIP and down-regulated transcripts were highly comparable (Fig. 7a, b), suggesting that miR-100 interacts with AGO2 to regulate targets in a way that is not explainable by seed base-pairing. Accordingly, CLASH[42] analysis of miR-100 and miR-125b-target chimeras showed that only 13% of the targets recognized by miR-125b are not explained

by seed complementarity, whereas this percentage rose to 71% for miR-100 (Supplementary Fig. 11b).

## miR-100 and miR-125b regulate common pathways in PDAC.

Ingenuity Pathway Analysis (IPA) of genes enriched for direct targets of miR-100 and miR-125b (Fig. 7a, b) showed that these two miRNAs regulate similar pathways ranging from p53 signaling, DNA repair and apoptosis (Fig. 7c–f), all of which are crucial for PDAC progression and metastasis[1]. Activation Z-score IPA values indicated that miR-125b significantly down-regulates p53 pathway, apoptosis and CHK proteins in cell cycle checkpoints in response to DNA damage, whereas miR-100 mainly down-regulates CHK proteins in cell cycle checkpoints in response to DNA damage, but also p53 and BRCA1 DNA repair signaling and apoptosis (Fig. 7c, d). Accordingly, the overlap between the transcripts regulated by both miRNAs was very significant and over 7-fold higher than expected by chance ($P < 2.2e-16$, exact hypergeometric probability) (Supplementary Fig. 11c). Targeting many of the same transcripts gives these miRNAs additional regulatory power, providing a means for double-bound targets to be more strongly repressed than others in order to maintain an integrated cellular response[41]. We summarized the interactions based on RIP-USE results (Fig. 6), between these miRNAs and the genes belonging to these pathways (Fig. 7e,f and Supplementary Data 4).

To evaluate the nature of the perturbations exerted by miR-100 or miR-125b, we selected both up- and down-regulated genes derived from the overexpression of each of these miRNAs (Z-score < −1.5 and Z-score > 1.5, Supplementary Data 5) and performed separate IPA analyses (Supplementary Fig. 11d,e and Supplementary Data 6). Combining the effects of these two miRNAs, we again showed that both significantly regulate common pathways, with the most important being CHK proteins in cell cycle checkpoints in response to DNA damage, p53 signaling, and apoptosis (Supplementary Fig. 11d and Supplementary Data 6). Comparison analysis combined with IPA Z-score values indicated that both miRNAs strongly down-regulate p38 MAPK, PTEN, p53 signaling, and apoptosis, yet up-regulate PI3K-AKT signaling and actin nucleation by ARP–WASP complex (Supplementary Fig. 11e and Supplementary Data 6), providing a rationale for why PDAC cells overexpressing these two miRNAs become more motile.

## miR-100 and miR-125b targets are repressed in mesenchymal cells.

Remarkably, the epithelial BxPC-3 cells express low amounts of miR-100 and miR-125b, whereas the mesenchymal-like metastatic S2-007 cells express very high levels (Supplementary Data 7). In fact miR-100 and miR-125b are the highest expressed miRNAs in S2-007, indicating that they are important drivers of PDAC aggressiveness. To demonstrate the impact of this change in expression, we performed RNA-seq of BxPC-3 and S2-007 (Supplementary Data 8). Notably, transcripts down-regulated in S2-007 versus Bx-PC3 cells were significantly enriched in miR-125b and miR-100 targets (Supplementary Fig. 12a, b). We found that 123 genes with seeds for miR-125b were down-regulated in S2-007 cells and overlapped with genes regulated by miR-125b in PANC-1 cells (Fig. 7a and Supplementary Fig. 12c). These 123 genes were still enriched for apoptosis, tight junction, and p53 pathways (Supplementary Fig. 12d). The number of overlapped genes for miR-100 with seeds was too low to have a reliable pathway enrichment analysis, probably due to the low amount of these consensus sites in the human transcriptome (Supplementary Fig. 10b, d and Supplementary Fig. 11a).

## Discussion

TGF-β can promote EMT by inducing SNAI1 and ZEB1/2, which in turn repress the transcription of the adherent junction gene CDH1[6]. Remarkably the miRNAs regulated during the TGF-β response in PDAC were previously unknown.

Herein, we show that TGF-β, via SMAD TFs, induces transcription of MIR100HG, which derives miR-100, let-7a, and miR-125b, but surprisingly only miR-100 and miR-125b are up-regulated (Fig. 8).

Remarkably, we found that inhibition of miR-125b or miR-100 affects the ability of TGF-β to induce cell motility, and the capacity of TGF-β to promote spindle-shaped cells, indicating that these miRNAs are important effectors of TGF-β. Interestingly, only miR-125b reverts the capacity of TGF-β induced tumourigenesis in vitro and in vivo, suggesting that miR-125b represents the most important effector of TGF-β-mediated tumourigenesis from those produced by MIR100HG. The lack of let-7a stimulation appears to result from TGF-β's induction of LIN28B which post-transcriptionally down-regulates let-7a levels (Fig. 8). To the best of our knowledge, a similar system that includes the up-regulation of a primary multi-cistronic miRNA-transcript, followed by the post-transcriptional inhibition of specific miRNAs within it, has not been demonstrated before, and we propose here that such regulation is important for the TGF-β response in PDAC. Interestingly, the correlations between both miRNAs with let-7a in clinical samples with low LIN28B levels are much weaker than the correlation between miR-125b and miR-100 themselves. This may be because identical let-7a sequences are produced from three different genomic locations that may be differentially regulated in different PDACs, thereby affecting correlation with members of MIR100HG lncRNA. Alternatively, other known or unknown let-7a post-transcriptional regulators maybe differentially expressed in tumor specimens affecting these correlations.

To find and characterize targets regulated by these miRNAs, we introduced a novel method based on ectopic modulation of miRNAs in cultured cells, followed by integration of AGO2-RIP-seq, RNA-seq, and sequence enrichment analysis, which we called RIP-USE. Although AGO2-RIP-seq has previously been used to identify miRNA targets[43] this method has never integrated RNA-seq with a detailed motif enrichment analysis, followed by cumulative distribution for validation of discovered miRNA–target interactions. RIP-USE can be applied to even poorly expressed miRNAs and for multiple cell lines, because it is based on ectopic modulation of the miRNA of choice, identifying targets that are also down-regulated. We propose that the integration of RNA-seq within this method would permit to statistically test the validity of canonical/noncanonical-mediated repression by performing cumulative distribution analyses. Importantly, RIP-USE, supported by a CLASH study[42], indicates that miR-100 uses AGO2 to regulate a much higher number of targets than the few that have complementary canonical or noncanonical seeds in their 3′UTRs, even without interacting with defined complementary sites in their seed regions.

Following activation by TGF-β, miR-100, and miR-125b target and down-regulate several genes belonging to the same pathways, which are all important during PDAC progression (Fig. 8). We hypothesize that the concurrent inhibition of these transcripts, by these two miRNAs, confers robustness to the TGF-β regulated processes. IPA analysis of the targets shows that both miR-125b and miR-100 significantly inhibit p53 signaling and apoptosis. Accordingly, we demonstrate that antagonizing both miRNAs increases sensitivity to GEM promoting apoptosis, therefore suggesting the potential usefulness of this as a clinical strategy to enhance GEM activity in PDAC.

Genes that are up- and down-regulated by these two miRNAs seem to be involved in a remarkable number of common pathways (Fig. 8). IPA Z-score values indicate that they both down-regulate genes and pathways normally associated with good cancer prognosis, such as p53, PTEN, and p38 MAPK signaling[44], and conversely up-regulate oncogenic and metastatic signaling such as PI3K/AKT and actin nucleation by ARP–WASP complex.

In summary, we describe here that miR-100 and miR-125b are both induced in PDAC by TGF-β signaling and cooperate to regulate similar pathways to promote stemness, EMT and tumourigenesis (Fig. 8). We propose that either single or concurrent inhibition of these miRNAs in conjunction with GEM could be considered as a potential adjuvant strategy for controlling PDAC, especially if patients have tumors overexpressing these miRNAs.

## Methods

**Cell culture**. BxPC-3, PANC-1, and COLO357 cells were obtained from American Type Culture Collection. S2-007 and S2-028 cells were obtained from Prof. Thomas Gress (University Hospital Marburg, Marburg, Germany). PANC-1 stably expressing TGF-β1 or empty vector were obtained from Prof Matthias Löhr and Dr Rainer Heuchel (Karolinska University Hospital, Stockholm, Sweden). CHX45 cells were obtained from Dr Bruno Sainz Jr (Department of Biochemistry, Universidad Autónoma de Madrid, Madrid, Spain)[33]. BxPC-3, COLO357, CHX45 and the two primary cell cultures from PDAC patients (LPC006 and LPC167)[45] were grown in RPMI-1640 medium supplemented with 10% fetal calf serum, 2 mM L-glutamine, 100 U ml$^{-1}$ penicillin, and 100 mg ml$^{-1}$ streptomycin. PANC-1, S2-007, and S2-028 were maintained in Dulbecco's modified Eagle medium supplemented with 10% fetal calf serum, 2 mM L-glutamine, 100 U ml$^{-1}$ penicillin, and 100 mg ml$^{-1}$ streptomycin. PANC-1 and S2-007 stable miRZip lines were expanded in puromycin (Gibco). All cells lines were tested monthly for mycoplasma contamination (MycoAlert, Lonza).

**Transfections and cell treatments**. Mimics, inhibitors, and Silencer Select siRNAs were purchased from Ambion. Mimics and inhibitors were transfected into cells using HiPerFect Transfection Reagent (Qiagen, Crawley, UK) whilst Silencer Select siRNAs were transfected using Lipofectamine RNAiMAX (Invitrogen) following the manufacturer's recommended protocol. Unless otherwise specified 5 nM of mimics, 100 nM of anti-miRs or 40 nM of siRNAs were transfected for 48 h. For experiments of siRNA co-transfection, 20 nM of each siRNA was used. Mimics: negative control #1 mimic (AM17110), pre-miR-100 (PM10188), pre-miR-125b (PM10148). Inhibitors: anti-miR negative control #1 (AM17010), anti-miR-100 (AM10188), anti-miR-125b (AM10148). siRNAs: negative control #2 (4390846), siLIN28B (4392420 s52479), siSMAD2 (4392420 s8397), siSMAD3 (4392420 s8400). For experiments of miRNA overexpression for 12 days, cells were transfected with the miRNA mimics (5 nM) and every 3 days cells were split and re-transfected with additional miRNAs mimics. For TGF-β treatments, PANC-1 cells were treated with 5 nM TGF-β (R&D Systems) for the indicated time. SB431542 (Sigma) was used at 2.5 μg ml$^{-1}$ in combination with TGF-β for 24 h. Gemcitabine (GEM) was used at 1 μM for 24 h.

**Generation of miR-Zip stable lines**. PANC-1 and S2-007 miR-100 and miR-125b knockdown stable lines were generated using miRZip$^{TM}$ lentivector-based anti-microRNAs technology (System Biosciences), following the manufacturer's protocol. miRZip lentiviral vectors stably inhibit the miRNA of interest by expressing a single-stranded shRNA which is recognized by DICER and processed to generate functional anti-miRNAs for the target miRNA. miRZip100 (Cat# MZIP100-PA-1), miRZip125b (Cat# MZIP125b-PA-1) or miRZip control (Cat# MZIP000-PA-1) were packaged using the pPACKH1 lentivector packaging kit (LV500A-1, SBI) in 293TN produced line (LV900A-1, SBI) and transduced into PANC-1 or S2-007 cells. Cells were selected with puromycin (1.6 μg ml$^{-1}$) for 7 days. The selected pool was then subjected to FACS sorting for GFP expression and 1 cell per well was seeded in a 96 well plate. Clones were left to grow and subsequently screened using the Cells-to-Cts protocol followed by QuantiMir RT kit (RA420A-1, SBI) using custom designed probes for Zip-100 and Zip-125b. The clones with the highest Zip-100 or Zip-125b expression were selected for phenotypic experiments. For in vivo metastasis assay S2-007 clones were transduced with lentiviral particles carrying red-shifted *Luciola italica* luciferase transgene (RediFect Red-Fluc-Puromycin, PerkinElmer). To assess the effect of TGF-β in vivo when miR-100 or miR-125b are inhibited, PANC-1 miR-100 Zip and miR-125b Zip clones were stably transduced with a lentiviral vector carrying TGFB-1 ORF (Cat# EX-Z5895-Lv151, GeneCopoeia$^{TM}$) or vector control (Cat# EX-NEG-Lv151, GeneCopoeia$^{TM}$). Cells were selected using G418 (800 μg ml$^{-1}$) for 7 days.

**CRISPR-Cas9-mediated KO lines**. All sgRNAs used for CRISPR-mediated PANC-1 KO lines were designed using the CRISPR design tool (http://crispr.mit.edu/). To generate mir-100 and mir-125b KO clones, pairs of sgRNAs were chosen within the miRNA genomic locus (see Fig. 4a) with the aim of disrupting part or the entire miRNA locus. For deletion of MIR100HG promoter region (MIR100HGΔP), pairs of sgRNAs were chosen to remove the SMAD2/3 peaks predicted by MACS2 from our ChIP-seq experiments. Finally, to generate LIN28B KO clones a single sgRNA targeting the genomic region downstream of the AUG translation start site codon was used. Oligonucleotides containing sgRNA sequences were cloned into lentiCRISPRv2 vector (a gift from Feng Zhang; Addgene #52961[46]) following the lentiCRISPRv2 and lentiGuide oligo cloning protocol deposited by the Zhang lab on Addgene. Oligonucleotides sequences are provided in Table 1. For LIN28B, miR-100 and miR-125b KO clones, lenti-CRISPRv2 containing the target sgRNA sequences were transfected in PANC-1 cells using Lipofectamine LTX (Invitrogen). At 24 h post-transfection, medium was replaced and cells were selected with 10 μg mL$^{-1}$ of puromycin (Gibco). After 48 h of antibiotic selection cells were left to grow and validation of CRISPR-Cas9 genome editing was assessed by Sanger sequencing of the PCR products of interest. Primers used for PCR amplification of genomic loci edited by CRISPR-Cas9 are indicated in Table 2. Cells with the desire genome editing were FACS sorted into 96 well plates as single cells and clones were left to grow for 7 days. Selected clones were further validated by Sanger sequencing and KO of the gene of interest was confirmed by RT-qPCR in case of miR-100 and miR-125b or western blot for LIN28B. The MIR100HGΔP line was generated by lentiviral infection in PANC-1 cells. Briefly, lentiCRISPRv2 containing the target sgRNA sequences were co-transfected with packaging plasmids pMD2.G and psPAX2 (both gifts from Didier Trono; Addgene plasmids #12259 and #12260) in HEK293T cells using Lipofectamine LTX (Invitrogen). Medium was refreshed after 8 h and lentivirus-containing medium was collected 60 h after transfection. PANC-1 cells were infected with lentivirus supernatant supplemented with 8 μg ml$^{-1}$ polybrene (Sigma). At 24 h post-infection medium was refreshed and cells were selected with 10 μg ml$^{-1}$ of puromycin for 48 h. Cells were left to grow and deletion of the promoter region was confirmed by Sanger sequencing.

**RNA isolation and RT-qPCR assays**. Total RNA from cultured cells was extracted using TRI Reagent (Sigma) following the manufacturer's instructions including DNase I treatment. qRT-PCR of mature miRNAs was performed using TaqMan Small RNA Assays (Applied Biosystems) with assays for hsa-miR-100 (000437), hsa-miR-125b (000449), hsa-let-7a (000377), has-miR-200a (000502). Small nuclear RNA U44 (001094), U47 (001223), or U6 (001973) were used as endogenous controls. For gene expression, cDNA was synthesized from 1 μg of purified DNase-treated RNA using RevertAid$^{TM}$ M-MuLV reverse transcriptase and random hexamer primers (Thermo Scientific), according to the manufacturer's protocols. We performed qRT-PCR on a StepOne$^{TM}$ Real-Time PCR System using Fast SYBR® Green Master Mix (both from Applied Biosystems). The primer sequences used are reported in Table 2.

**Sphere formation assay**. BxPC-3, PANC-1, PANC-1 ZiP stable lines, PANC-1 CRISPR-Cas9 KO lines or PANC-1 TGF-β stables were plated as single cell suspension in 6 well ultra-low attachment plates (Corning) at a density of $2 \times 10^3$ cells/well. Cells were grown in serum free DMEM/F12 medium (Gibco) supplemented with B27 (1:50, Gibco) and 20 ng ml$^{-1}$ basic fibroblast grown factor (bFGF, Biolegend). Tumor spheres were counted at day 7 when typically reached a size of >75 μm. Sphere formation efficiency (SFE) was calculated as the number of spheres formed at day 7 divided by the number of cells seeded and expressed as a percentage.

### Table 1 Oligonucleotides for sgRNA cloning

| sgRNA | Sequence |
| --- | --- |
| mir-100 sgRNA1-F | CACCG**ATCTACGGGTTTGTGGCAAC** |
| mir-100 sgRNA1-R | AAAC**GTTGCCACAAACCCGTAGAT**C |
| mir-100 sgRNA2-F | CACCG**TTAGGCAATCTCACGGACC** |
| mir-100 sgRNA2-R | AAAC**GGTCCGTGAGATTGCCTAA**C |
| mir-125b sgRNA1-F | CACCG**ACCGTTTAAATCCACGGGTT** |
| mir-125b sgRNA1-R | AAAC**AACCCGTGGATTTAAACGGT**C |
| mir-125b sgRNA2-F | CACCG**CGAGTCGTGCTTTTGCATCC** |
| mir-125b sgRNA2-R | AAAC**GGATGCAAAAGCACGACTCG**C |
| MIR100HGΔP sgRNA1-F | CACCG**CAAGAGTGTAAAGACCCCGA** |
| MIR100HGΔP sgRNA1-R | AAAC**TCGGGGTCTTTACACTCTTG**C |
| MIR100HGΔP sgRNA2-F | CACCG**GAGCATACGTGTCCCCAT** |
| MIR100HGΔP sgRNA2-R | AAAC**GATGGGGACACGTATGCTCC**C |
| LIN28B sgRNA-F | CACCG**CCGTGGGGCAACATGGCCGA** |
| LIN28bB sgRNA-R | AAAC**TCGGCCATGTTGCCCCACGG**C |

The 20 bp sgRNA sequence is highlighted in bold

**Table 2 PCR primers used in this study**

| Primer | Sequence |
| --- | --- |
| miR-100 genomic-F | AAAGTGGAAACCAAGGGAAGCAC |
| miR-100 genomic-R | CTCATTCATTTCAGGACAAAAGGTC |
| miR-125b genomic-F | GAAGAAATACCATACCACCTGTT |
| miR-125b genomic-R | GTCACCTGATCCCATCTAACAAT |
| LIN28B genomic-F | TAGATTGATGCAGAAGATCACTCC |
| LIN28B genomic-R | AAGTTGTGAATCAGTGTGGG |
| MIR100HGΔP genomic-F | TTTCCATGTAAGAATGGTCTCC |
| MIR100HGΔP genomic-R | TTCATTCTATTTCCTGAAGCTGGG |
| pre-miR-100-F | AACCCGTAGATCCGAACTTG |
| pre-miR-100-R | TACCTATAGATACAAGCTTGTGCG |
| pre-miR-125b-F | GTCCCTGAGACCCTAACTTG |
| pre-miR-125b-R | AGCCTAACCCGTGGATTT |
| LIN28B-F | TTAGGAAGTGAAAGAAGACCCA |
| LIN28B-R | ACCACAGTTGTAGCATCTATCT |
| CDH1-F | CCCACCACGTACAAGGGTC |
| CDH1-R | CTGGGGTATTGGGGGCATC |
| SNAI1-F | GAGGCGGTGGCAGACTAG |
| SNAI1-R | GACACATCGGTCAGACCAG |
| CD133-F | TCGACAATGTAACTCAGCGT |
| CD133-R | CCCAGCCACCAGTATGAAT |
| ANKRD28-F | GGACATGGTGAGATGGTCAA |
| ANKRD28-R | CCAATGGATAGCACGCCT |
| CPEB2-F | GAGCAGAGCATGATCCTCTT |
| CPEB2-R | AGAGGGAAGAACGACCATTT |
| PXN-F | CCCATCCTGGATAAAGTGGT |
| PXN-R | GCACAGAAGAAGTGTTCAGG |
| NF2-F | TCCAGCTATGTATCGGGGAAC |
| NF2-R | CCGCTCCATCTGCTTTCTA |
| NUMB-F | CCAGTCGTCCACATCAGT |
| NUMB-R | ACAGATGTGCATTCCTCTTGA |
| MIR100HG-F | ACACAGACTTGTCTTTGGACA |
| MIR100HG-R | AAACCTGCTTCCATCTTGTTAG |
| U6-F | CTCGCTTCGGCAGCACA |
| U6-R | AACGCTTCACGAATTTGCGT |
| GAPDH-F | TGAAGGTCGGAGTCAACGGATTT |
| GAPDH-R | GCCATGGAATTTGCCATGGGTGG |

**Western blotting**. Whole cell lysates were prepared in RIPA buffer (Sigma) and quantified using Bradford Protein Assay (Bio-Rad). Twenty micrograms of lysates were resolved on Bolt® 4–12% Bis-Tris Plus gels using Bolt™ MOPS SDS Running buffer and transferred to a Hybond C super nitrocellulose membrane (GE Healthcare). After blocking to prevent non-specific binding in 5% milk in PBST for 1 h at room temperature, membranes were incubated with the specific primary antibodies overnight at 4 °C. The following primary antibodies were used: E-cadherin (Takara Bio Inc., M106, clone HECD-1, 1:1000 dilution), Occludin (Cell Signaling, #5446, 1:1000 dilution), Vimentin (Cell Signaling, #5741, 1:3000 dilution), ZEB1 (Santa Cruz, sc-25388, 1:500 dilution), SNAI1 (Santa Cruz, sc-28199, 1:500 dilution), LIN28B (Cell Signaling, #4196, 1:1000 dilution), LIN28A (Cell Signaling, #3978, 1:1000 dilution), SMAD2/3 (Cell Signaling, #8685, 1:1000 dilution), β-actin (Abcam, ab8227, 1:200,000 dilution), GAPDH (Santa Cruz, sc-137179, 1:10,000 dilution). Following incubation with the specific HRP-conjugated antibody (Dako, #P0447 or #P0448, 1:2,500 dilution), chemiluminescence signal was detected using Amersham™ ECL™ Western blotting detection reagents (GE Healthcare) and Amersham Hyperfilm™ ECL (GE Healthcare). Uncropped scans of the most important blots are shown in Supplementary Fig. 13.

**Immunofluorescence staining**. Cells grown on glass coverslips were fixed in 4% paraformaldehyde for 10 min at room temperature. Cells were washed twice with PBS (15 min each) and permeabilised with 0.3% (v/v) Triton X-100 for 10 min. Blocking buffer (1% w/v BSA, 2% v/v FCS, 5% v/v goat serum in PBS) was added for 30 min and cells were then stained with anti-E-cadherin antibody (Takara Bio Inc., M106, clone HECD-1, 1:1000 dilution) and Alexa Fluor® 555 (Invitrogen, A-21422, 1:500 dilution). F-actin was detected with phalloidin-Alexa Fluor 488® (Invitrogen, A12379, 1:500 dilution) and nuclei were visualized using TO-PRO®-3 (Invitrogen, T3605, 1:1000 dilution). Images were acquired with a Zeiss LSM 510 META confocal microscope (Carl Zeiss Ltd).

**Wound-healing migration assays**. Confluent cells were scratch-wounded using a 20 µl pipette tip and cell debris were removed by washing with PBS. Phase-contrast

images were taken at the indicated time points at specific wound sites using EVOS microscope with a 10 × objective. For time-lapse experiments, cell migration was assessed by phase-contrast videomicroscopy with sequences taken every 30 min for 24 h after scratching using Axiovert 100 MetaMorph Microscope (Carl Zeiss Ltd) fitted with a humidified 37 °C incubation chamber. For wound-healing assay in adenocarcinoma primary lines LPC006 and LPC067, cells were transfected with 100 nM of anti-miR-NC, anti-miR-100 or anti-miR-125b and seeded in 96 well plates (30 × 10⁴ cells/well), where artificial wounds were made using a pipette tip. The ability of the cells to migrate was evaluated by comparing the pixels of the wound tracks in the images taken at the beginning of the exposure (time 0) with those taken 2 hourly for 8 h and then finally at 16 h. Migration was assessed using the LeicaDMI300B (Leica) migration station integrated with the Scratch-Assay 6.1 software (Digital-Cell Imaging Labs, Keerbergen, Belgium).

**Flow cytometry**. To identify CD133 positive population, 1 × 10⁶ S2-007 Zip cells were trypsinized, washed with PBS and stained with antiCD133/1-APC (Miltenyi Biotec) or appropriate isotype-matched control antibody for 30 min at 4 °C. The labelled cells were washed in PBS and subjected to flow cytometry on a FACS Canto II (BD Bioscieces). Data were analyzed with FlowJo v8.8.9 software.

**Apoptosis assay**. LPC006 and LPC067 cells were transfected with 100 nM of anti-miR-NC, anti-miR-100, or anti-miR-125b and the next day were treated with 1 µM gemcitabine (GEM) chemotherapy for 24 h. Following treatments, cells were washed twice with PBS and fixed in 4% PFA for 15 min. Cells were then resuspended in a solution containing 8 µg ml⁻¹ bisbenzimide HCl and incubated for 15 min. Cells were spotted on glass slides and were examined by fluorescence microscopy (Leica, Wetzlar, Germany). A total of 200 cells from randomly chosen microscopic fields were counted, and the percentage of cells displaying chromatin condensation and nuclear fragmentation relative to the total number of counted cells (apoptotic index) was calculated. The average percentage of apoptosis induced by GEM and anti-miR-NC (negative control) was 19% and 12% in the LPC006 and LPC067 cells, respectively. These values were set as 100%, in order to show the difference after treating with the other anti-miRs more clearly. Apoptosis induction <20% indicates that both cell lines are not very sensitive to GEM.

**Luciferase reporter assays**. PANC-1 cells were seeded onto 24 well plates at a density of 50 × 10⁴ cells/well in antibiotic-free medium. Twenty-four hours later, cells were co-transfected with pre-miR-100 or pre-miR-125b or negative control (100 nM) together with the 3′UTR of relevant genes reporter constructs (pLightSwitch_3UTR GoClone vectors, SwitchGear Genomics) at 100 ng/well using Lipofectamine2000 (Invitrogen). After 24 h, cells were washed with PBS and 50 µl of passive lysis buffer (Promega) was added to each well. After a 20-min incubation lysates were transferred to an optical quality 96 well plate (OptiPlate, PerkinElmer) and luciferase activity was measured using the LightSwitch Assay System (SwitchGear Genomics) as directed by the manufacturer. The resulted light emission was read using the Infinite M200 plate reader (Tecan) and the mean luciferase activity for each precursor miRNA is shown relative to the mean for the negative control.

**miRNA nCounter profiling**. The Nanostring nCounter Human miRNA Expression Array (http://www.nanostring.com/) was used to obtain miRNA expression profiles[47]. nCounter miRNA sample preparation was performed according to the manufacturer's instructions (NanoString Technologies). Differential expression analysis between different cell lines and PANC-1 cells treated with TGF-β or vehicle was performed in R, using nCounter raw values as input for DESeq2 analysis (http://bioconductor.org/packages/release/bioc/html/DESeq2.html).

**AGO2 RNA immunoprecipitation**. PANC-1 cells were seeded in 15 cm dishes (5 × 10⁶ cells/dish; 3 dishes were used for each antibody tested) and transfected with 0.5 nM of pre-NC, pre-miR-100 and pre-miR-125b for 24 h. AGO2-RIP was carried out as previously described by our group[48]. Briefly, following transfection cells were washed in cold PBS, scraped in PBS and collected by centrifugation. Pellets were then resuspended in lysis buffer (20 mM Tris-HCl pH7.5, 150 mM KCl, 0.5% NP40, 2 mM EDTA, 1 mM DTT, 0.5 mM NaF, 160 U ml⁻¹ RNAsin and protease and phosphates inhibitors) and pre-cleared with Protein G sepharose beads (Sigma) for 2 h at 4 °C. Part of cleared lysates (10%) was used as input and the remainder were incubated with Protein G sepharose beads conjugated with anti-AGO2 (11A9, SAB4200085, Sigma-Aldrich) or anti-IgG (Sigma-Aldrich) for 4 h at 4 °C. After washing, 10 µl of the immunoprecipitate was kept for western blot analysis and the remainder was treated with DNAse I and proteinase K for 20 min at room temperature. RNA was extracted using phenol/chloroform and and ethanol/sodium acetate precipitation. RNA was then quantified using Nanodrop.

**Chromatin immunoprecipitation**. PANC-1 cells were treated with 5 nM TGF-β for 24 h and chromatin immunoprecipitation (ChIP) was performed as follows[49]: briefly cells were crosslinked with 1% formaldehyde, chromatin was prepared and incubated with 10 µg of SMAD2/3 antibody (R&D, AF3797) and 100 µl of Dynabeads Protein A (10002D; Invitrogen) overnight at 4 °C. The immune-precipitated

complex was washed with RIPA buffer and TE buffer, followed by de-crosslinking for 16 h. DNA was then treated with RNase and Proteinase K at 1 mg ml$^{-1}$ and purified with phenol-chloroform and sodium chloride precipitation. For ChIP-PCR the primer sequences are as follow: SNAI1 (Forward: 5′-CGCTCCGTAAA-CACTGGAT-3′; Reverse: 5′-GAAGCGAGGAAAGGGACAC-3′), MIR100HG (Forward: 5′-AGCAAACACATTTCAGGCAGT-3′; Reverse: 5′-GGCTACCT-GACTGATGAGTG-3′), HBB (Forward: 5′-GCTTCTGACACAACTGTGTTCAC-3′; Reverse: 5′-CACCAACTTCATCCACGTTCACC-3′).

**ChIP-seq library preparation and bioinformatics analysis**. ChIP-seq libraries were constructed using NEBNext Ultra II DNA library prep kit for Illumina kit (NEB), according to the manufacturer's protocols. Ten nanograms of DNA was used for library preparation. Single-end reads of 50 nt in length were generated using a HiSeq 2000 instrument (Illumina). Sequences were aligned to the human reference genome (assembly hg19, February 2009) using Bowtie 1.0. and peak calling was performed using MACS v1.4 using default settings. ChIP-seq data of CTCF, H3K4me3, and H3K27ac from PANC-1 or Pancreata were obtained from ENCODE and plotted using IGV genome browser.

**Hi-C analyses**. Hi-C data of PANC-1 comes from ENCODE. Analyses and graphs were performed using online tools from http://promoter.bx.psu.edu/hi-c/.

**RNA-seq and AGO2-RIP-seq library preparation**. RNA libraries for RNA-seq and AGO2-RIP-seq were prepared with TruSeq RNA Library Prep Kit v2 (Illumina), according to the manufacturer's protocols. Paired-end sequences (reads) of 100 nt in length were then generated using a HiSeq 2000 instrument (Illumina).

**Processing of RNA-seq and AGO2-RIP-seq data**. The quality of the reads contained in the fastq files obtained at the end of the sequencing was assessed using FastQC version 0.10.1 (http://www.bioinformatics.babraham.ac.uk/projects/fastqc/). The reads from the fastq files, for each sample, were then mapped on the reference human genome, version hg19, obtained from the University of California Santa Cruz (UCSC) genome browser (https://genome.ucsc.edu/) by using TopHat. For isoform level analysis (miRNA target identification) RPKM normalized values were produced with Partek Genomic Suit software (Partek Inc) using the bam files attained after the TopHat runs, as input. For gene level analysis (TGF-β treatment) raw counts were produced using htseq version 0.6.1 (http://www-huber.embl.de/HTSeq/) with human RefSeq annotation and used for differential expression analysis with DESeq2 from the Bioconductor (https://www.bioconductor.org/).

**RIP followed by Unbiased Sequence Enrichment (RIP-USE)**. We developed RIP-USE for miRNA-target identification in order to identify canonical and non-canonical targets for miR-100 and miR-125b. It integrates AGO2-RIP-seq with RNA-seq and unbiased motif enrichment analysis to identify enriched motifs complementary to any part of the miRNAs enriched in the transcripts loaded onto AGO2 upon expression of miR-125b or miR-100 in cell lines. The function of these motifs in regulating targets through miRNA interaction was then tested by performing cumulative distribution analyses comparing the global expression of transcripts containing identified sites versus transcripts without them, upon miRNA expression. It consists of different steps (Fig. 6a): (1) miRNA over-expression in cell lines, (2) AGO2-RIP-seq of the cells overexpressing the miRNA of interest or a negative control (n.c.), (3) RNA-seq of the cells overexpressing the miRNA of interest or a negative control (n.c.). After mapping of the sequencing reads followed by gene expression analysis (4) the transcripts are then sorted from the most enriched to the least enriched in AGO2 for AGO2-RIP-seq, as well as from the least down-regulated to most up-regulated for RNA-seq. Considering that usually the region of the miRNAs that base pairs with their targets correspond to a 6–8 mer located in the 5' part called the 'seed'[50] the genes enriched for AGO2 and the ones down-regulated after the expression of the miRNA of interest should be enriched of words 6–8 bases long complementary (canonical pairing) or partly complementary (noncanonical pairing) with the seed of the overexpressed miRNAs. Considering this principle, (5) to find bona fide targets of the overexpressed miRNAs we used tools that unbiasedly retrieve enriched words 6–8 bases long within selected regions of sorted transcripts[38,40,51] for both AGO2-RIP and RNA-seq. We evaluated whether (6) words representing noncanonical interaction derived from regions of enriched transcripts onto AGO2 for RIP-seq overlap with the ones from regions of down-regulated transcripts for RNA-seq. Finally (7) we validated whether the transcripts containing these 6–8mers are actually regulated by the miRNAs, evaluating whether transcripts containing those words were significantly down-regulated compared to transcripts without those words in RNA-seq upon the overexpression of the miRNAs, using cumulative distribution analysis (Fig. 6b,c).

**Pathway analyses**. Pathway analyses were performed using IPA (Qiagen, http://www.ingenuity.com/products/ipa). Genes from the overlap between the most enriched transcripts in AGO2-RIP-seq ($n = 4000$) and the top down-regulated transcripts in RNA-seq (1300) were used as gene sets for the analysis (Fig. 7a,b). Expression values (Log2FC) were also including to obtain the IPA Z-score that

infers onto activation (positive Z score) or inhibition (negative Z score) of the significant enriched pathways. Networks of top enriched IPA pathways were built using Cytoscape software v3.4 (http://www.cytoscape.org/). For IPA comparison analysis, genes that were up and down-regulated (Z-score > 1.5 and Z-score < −1.5) were analyzed.

**Analysis of mRNA and miRNA expression profiling from The Cancer Genome Atlas (TCGA) database**. Level 3 data of miRNA and mRNA expression profiling data from pancreatic adenocarcinoma samples (PPAD-TCGA) were downloaded from the TCGA database (https://portal.gdc.cancer.gov/projects/TCGA-PAAD) and used to develop gene and miRNA expression matrices. Pearson correlation analyses between miRNAs as well as between miRNAs and mRNAs of interest were calculated and graphed using R (https://www.r-project.org/). We considered low LIN28B level samples to be those that express less than 0.5 RSMEs (https://bmcbioinformatics.biomedcentral.com/articles/10.1186/1471-2105-12-323).

**In vivo tumorigenicity and metastasis assays**. For tumorigenesis assays, serial dilutions of S2-007 Zip cells diluted in 50 µl PBS were injected subcutaneously in both flanks of 4- to 5-week-old female athymic nude mice (Charles River Laboratories). Control cells were inoculated to the right flank and Zip100 or Zip125b cells to the left. Tumor growth was documented for 3 weeks. For TGF-β tumorigenesis assay, $2 \times 10^6$ PANC-1 Zip cells stably expressing TGF-β1 or empty vector diluted in 50 µl PBS were injected subcutaneously in both flanks of 4- to 5-week-old female athymic nude mice (Charles River Laboratories). Tumor growth was documented for 5 weeks. For metastasis assay, $25 \times 10^4$ S2-007 Zip cells stably expressing luciferase resuspended in 50 µl PBS were intrasplenically injected into 4- to 5-week-old female athymic nude mice (Charles River Laboratories). Seven days after injection the spleen was removed to avid growth of the tumor at the injection site. Tumor dissemination was monitored twice a week for 3 weeks by biolumi-nescence imaging using the IVIS Spectrum Imaging System (Caliper Life Sciences). Briefly, mice were injected intraperitoneally with 150 mg kg$^{-1}$ of luciferin (Promega, E1605) diluted at 15 mg mL$^{-1}$ in sterile PBS and imaged after ~15 min. The Living Image Software (Caliper Life Sciences) was used to quantify photon emission and regions of interest (ROI) was used to calculate the total photons/second (photons flux) emitted. The number of animals used in all experiments reflected the expected magnitude of response taking into account the variability observed in previous experiments. Mice were randomly allocated to each group. Investigators were not blinded. Animal experimental procedures were conducted under the UK Home Office Project License number PPL 70/8448.

**Laser capture microdissection (LCM) of tissues**. Samples of matched tissues (normal pancreas, $n = 8$; PDAC, $n = 20$; lymph-node metastasis, $n = 20$) were obtained from FFPE blocks of surgical specimens after pancreaticoduodenectomy (all resectable Stage IIB) at Hammersmith Hospital, UK. Cells were selectively isolated with the PALM laser capture microdissection (LCM) platform (Carl Zeiss Ltd., Cambridge, UK) according to the manufacturer's protocols. This was performed to allow confirmation of cell-type specific changes in miRNA expression[52]. Total RNA was subsequently extracted using the RNeasy FFPE Kit (Qiagen, Hilden, Germany) following the manufacturer's protocol. Informed consent was obtained from all patients and ethical approval was received from the Camden & Islington REC, London (09/H0722/77) in the UK.

**LNA-based microRNA in situ hybridization**. Patients: The aim of this experiment was to study expression of miR-100 and miR-125b in human PDAC tissue samples by LNA-based miRNA ISH. The experiment included formalin-fixed paraffin embedded (FFPE) tumor specimens from 100 PDAC patients (all resectable Stage IIB) arranged on 4 tissue-microarrays (TMA), each containing 25 patient sample. For each patient tumor there were 4 cores (1.5 mm diameter) in order to avoid intra-tumoral heterogeneity and best represent the tumor. Hematoxylin and eosin (H&E) staining was performed on all samples prior to processing for the ISH analysis in order to confirm tumor histology. Patients underwent surgery for PDAC at the University of Pisa, Italy during 2005–2010 and were closely followed-up. None of the patients received neo-adjuvant chemotherapy, but all received adjuvant chemotherapy. Complete clinicopathological, follow-up and recurrence data were available from a prospectively maintained database.

LNA probes: DNA oligonucleotides with approximately 30% Locked Nucleic Acid (LNA) substitutions[53] for the full length miRNA were used: miR-100-5p (predicted Tm ~ 85 °C; target sequence CACAAGTTCGGATCTACGGGTT; 32% LNA) and miR-125b-5p (predicted Tm ~ 85 °C; target sequence TCACAAGTTAGGGTCTCAGGGA; 27% LNA) (Exiqon, Vedbaek, Denmark). In addition, a probe specific for U6 snRNA (ACGAATTTGCGTGTCATCCTT; predicted Tm ~ 83 °C; 29% LNA) was used as positive control, and a 21-mer scrambled probe with a random sequence (TGTAACACGTCTATACGCCCA; predicted Tm ~ 87 °C; 33% LNA) having no known complementary sequence target among human transcripts performing MegaBLAST search at NCBI GenBank, was included as negative control (Exiqon, Vedbaek, Denmark). All LNA oligos were digoxigenin (DIG)-labeled at the 5′- and 3′-ends except the U6 probe, which was only 5′-end labeled.

In situ hybridization: Five μm-thick paraffin TMA sections were mounted on Super frost + glass slides and deparaffinized. ISH for miRNA detection was carried out using a miRCURY LNA miRNA ISH kit (Exiqon, Vedbaek, Denmark) as previously described[54] with few modifications. For optimization, miR-100 and miR-125b probe concentrations and proteolytic pre-treatment were evaluated on 4 separate full-size FFPE PDAC sections. A proteinase-K (PK) pre-treatment of 20 μg ml$^{-1}$ and probe concentration of 30 nM were chosen for subsequent TMA analyses.

Image analysis and quantification: Images were acquired using a 20× and 40× objectives with a Zeiss AxioScan. For miRNA quantification, the following histologically stained structures were differentiated: blue areas corresponding to the hybridization signal, red area corresponding to the red nuclear stain, purple areas corresponding to blue ISH signal overlaying nuclear red stain. Scoring of ISH was performed semi-quantitatively based on staining intensity: 0, absent; 1+, weak (visible at 200× magnification); 2+, moderate (visible at 100× magnification); 3+, strong (visible at 40× magnification). In addition, localization of miRNA signal in either the tumoral and stromal compartments was assessed. Thus, 3 parameters were obtained from image analysis and reflecting relative miRNA expression levels were considered: (1) the stained area, (2) the staining intensity and (3) the number of positive cells. This tissue slide-based miRNA ISH therefore provided qualitative (tumor compartment) and semi-quantitative (expression levels) data.

Semi-quantitative ISH score: In order to accurately describe the extent of ISH staining of a tumor core, the degree of ISH staining in each compartment (tumor or stromal cells) was measured for each miRNA. We captured the percentage of cells stained at each intensity level and then used an intensity scale ranging from 0 for no staining, to 3+ for the most intense staining. This information was used to calculate a variable ISH Score, more continuous than simply positive versus negative. This ISH score for each patient tumor or stromal compartment was calculated using the mean intensity plus % of cells of the 4 cores. The mean ISH score for all patients was then used to dichotomize patient cores into those with high or low miRNA expression in the cellular compartments.

Statistical analysis: MiRNA levels in tumor and stromal compartments were compared with the Student's *t*-test (2-tailed). Patients dying from PDAC after surgical resection were considered as "death events" and this was used to calculate the OS analysis. Date of death was obtained from the hospital records, family doctor or Italian Civil Registration. Patients who were categorized as having an event in the calculation of DFS experienced local tumor recurrence in the form of either local or distant metastasis. High and low miRNA ISH score, using the mean value as a cut-off, was then used to associate miRNA expression with survival outcomes following pancreaticoduodenectomy (PD) surgery (i.e. from date of surgery to date of death, or date of disease progression). Survival curves were constructed using the Kaplan-Meier method and differences in survival were compared by log-rank test. All statistical analyses were done using SPSS 20.0 (IBM, SPSS). *P* values ≤0.05 were considered significant.

**Data availability**. The accession number for the Nanostring, RNA-seq, RIP-seq, and ChiP-seq data reported in this paper is GEO: GSE88759. All other remaining data are available within the Article and Supplementary Files, or available from the authors upon request.

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

## Acknowledgements

The authors thank Action Against Cancer (AAC), Pancreatic Cancer UK (PCUK), The Academy of Medical Sciences, The Royal College of Surgeons of Edinburgh, The Colin McDavid Family Trust, No Surrender Cancer Trust (in memory of Jason Boas), Mr Alessandro Dusi, Cheryl Whitehead, BHM, Sir Douglas Myers, and The Ralph Bates Pancreatic Cancer Research Fund for funding this study. Financial support was provided by the Dutch Cancer Society, KWF#10401 grant to E.G., Italian Association for Cancer Research AIRC/Start-Up grant to E.G., Istituto Toscano Tumouri ITT-grant to N.F. and E.G., and the Regione Toscana "Progetto DIAMANTE"/FAS grant to N.F. and E.G. The authors thank Prof. Matthias Löhr and Dr Rainer Heuchel at Karolinska University Hospital, Stockholm, Sweden, for providing the TGF-β1 and empty vector expressing PANC-1 cells. This work used the computing resources of the UK MEDical BIOinformatics partnership—aggregation, integration, visualization, and analysis of large, complex data (UK MED-BIO) which is supported by the Medical Research Council [grant number MR/L01632X/1].

## Author contributions

L.C. conceived the project, supervised and designed all research. L.C. and S.O. wrote the manuscript. L.C. designed and conducted the bioinformatic analysis. J.S. provided necessary reagents. S.O. designed and performed the most of the experimental work, A.E. F., S.Z., J.K., A.deG., S.M.T., V.T.M.N., H.F., E.G., N.F., and Y.L. performed the experiments. L.M. supervised and analyzed the ChIP-seq study. T.M.G. provided the S2-007 and S2-028 cell lines. L.R.J. provided the clinical samples and other reagents. N.R.L. provided reagents and edited the manuscript. C.H. supervised the in vivo studies.

## Additional information

**Competing interests:** One of the authors, Y.L., is an editor on the staff of Nature Communications, but was not in any way involved in the journal review process. The other authors declare no competing interests.

