## [Peer Review File · Nature Communications]

Reviewers' comments:

Reviewer #1 (Remarks to the Author):

In this study, Ottaviani et al characterize the effect of TGF β signaling on specific microRNAs. While the results are provocative and point out a novel way in which TGF β signaling modulates the phenotype of PDAC cells, the study as written could be improved with attention to the following:

1. The authors correctly point out in their introduction that SMAD4 is a frequently mutated tumor suppressor gene in PDAC that acts in the TGF β pathway. Thus, one could hypothesize that the downstream mediators of TGF β signaling might differ between SMAD4 wildtype and mutant PDACs. It does not seem that the authors consider this possibility. Can they comment on possible differences in microRNA signaling between these two groups? Are the cell lines utilized in their experiments SMAD4 mutant or wildtype? Perhaps SMAD4 mutation is not important here since the authors suggest that the regulation is directly through SMAD2/3, but they should at least discuss this issue.
2. The authors use the terms EMT and CSC very loosely. In my opinion, they should be more precise in the phenotypes they are characterizing. For example, the authors claim that overexpression of their micro-RNAs induces EMT - as evidence they use morphology, expression and localization of 2 proteins, and motility in a wound healing assay. To this reviewer, it is acceptable to raise the possibility that these specific phenotypes are characteristic of EMT in the Discussion, but the results should not stretch the interpretation of these findings. Similarly, the authors claim that their microRNAs are important for "CSC formation" but then use a small number of very specific assays to assess this possibility. While the results are interesting and have implications about the functional role of the microRNAs, I do not agree that they are specifically indicative of the role of cancer stem cells. Such speculation is better left in the Discussion.
3. Since cancer stem cells are thought to represent a small proportion of cells in a primary tumor with a specific phenotype, it is unclear whether the reported cell line experiments can really address the role of the microRNAs in this phenotype. How can a cell line recapitulate the functional heterogeneity inherent in the cancer stem cell hypothesis?
4. The correlation of micro-RNA expression with OS is quite interesting. Can the authors also correlate with grade of differentiation of the primary tumors? Are those with high micro-RNA expression less differentiated/more mesenchymal in morphology on histological sections? This is important for the clinical relevance of the findings.
5. The authors suggest that these microRNAs might be therapeutic targets. Have any strategies been reported for targeting microRNAs in vivo? If so, these studies should be referenced. Otherwise, such a clinical application is a stretch, since targeting these is likely quite challenging.

Minor comments:

1. The following statement in the introduction is not accurate: "In PDAC, 95% of cases show mutational hyper-activation of KRAS and inactivation of the tumour suppressors TP53, P16/INK4A, and the TGF- β effector, SMAD4". 95% of PDACs have KRAS mutations, but the prevalence of tumor suppressor gene alterations is much lower - SMAD4 is altered in only ~50% of PDACs.
2. I think a word has been omitted from this heading: "TGF- β increases miR-100 and miR-125b inducing MIR100HG transcription through SMAD2/3" Should it say "BY inducing"? As written it suggests that miR-100 and miR-125b induce MIR100HG, which is not supported by the data.

Reviewer #2 (Remarks to the Author):

In this manuscript, Ottaviani and colleagues propose a novel mechanism through which TGF-beta

promotes EMT and cancer stem cell in pancreatic cancer.

According to this model, TGFbeta transcriptionally activates a the miR-100~let7a2 polycistronic miRNA cluster, which encodes for miR-100, miR-125b, and let-7a2. However, TGFbeta seem to also induce expression of LIN28 (an inhibitor of let-7 processing) and therefore while the levels of mature miR-100 and miR-125b increase, let-7a2 levels remain largely unchanged.

The selective upregulation of miR-100 and miR-125b, in turn leads to the direct and indirect modulation of a large set of genes, promoting EMT and stemness of pancreatic cancer cell lines.

The manuscript is clearly written, although the section on RIP-USE would benefit from some editing, and the topic is of substantial interest. The experiments are clearly described and the appropriate controls are included.

My major concern with this work is that the authors do not directly test the central claim of the paper: that induction of miR-100/125b by TGFb is important for EMT and cancer cell 'stemness'. To be more precise: in figure 1 they show that TGFbeta treatment of PANC1 cells leads to a rather modest (less than 2 fold) increase in miR-100b and miR-125b. although it is certainly possible that even a modest increase in a miRNA can have substantial phenotypic consequences, this need to be experimentally demonstrated.

To support their model the authors performed two key experiments:

a) ectopic expression of miR-100 and miR-125b in BxPC-3, PANC-1 and CHX45 cells. Although these experiments seem to show a partial induction of EMT (Fig 2Am, B), miR-100 and miR-125b are likely induced to much higher levels compared to what the authors report in response to TGFbeta treatment, and therefore they do not directly address the central hypothesis of the manuscript.

b) The second key experiment is using the miRZIP platform to inhibit miR-100 and miR-125b in pancreatic cancer cells expressing high levels of these miRNAs. Also in this case, the results show that inhibiting these miRNAs can impair cell migration (Fig 2D-F) and induce a more 'epithelial' morphology (Fig 2C), but bear little relevance to the model proposed by the investigators.

c) more convincing are the experiments reported in figure 4, in which miR-100 and miR-125b inhibitors are shown to impair the ability of TGFbeta to increase the sphere formation efficiency of PANC-1 cells.

Although these experiments certainly link miR-100 and miR-125b to EMT and CSCs, they do not directly test the ability of TGFbeta to induce EMT and CSCs in the via selective miR-100/miR125b induction. It must be pointed out that several papers have already linked miR-100 and miR-125b to EMT (for example PMID: 24586203, 24805183, 27383536) although a consensus on whether these miRNAs promote or antagonize EMT has not emerged yet.

In summary, the relevance and quality of this manuscript would be greatly improved if the authors included experiments directly testing the ability of TGFbeta to promote EMT and CSCs via miR-100/miR-125b induction. More specifically, I would recommend performing the following relatively straightforward experiment:

Using CRISPR-Cas9, the investigators should delete the two Smad2/3 binding sites in the MIR100HG promoter region (Figure 2a) in PANC-1 cells (or in another suitable model system). This should selectively impair the ability of TGFbeta to induce miR-100/miR-125b. Alternatively, the investigators could simply delete the two miRNA precursors.

They should then test the ability of TGFbeta to induce partial EMT and promote CSC in these cells compared to control cells with the intact locus. This simple experiment should conclusively test the central hypothesis of the manuscript, complementing the other data presented, and overcoming the intrinsic limitations of overexpression and miRZIP studies.

Additional points:

- 1) A western blot showing LIN28 induction in response to TGF beta should be included
- 2) the scatter plot shown in figure 2 are a bit confusing. It seems that the majority of samples have low levels of LIN28. The criteria for defining low or high LIN28 levels, as well the total number of samples in each category and the correlation test used should be specified. I couldn't find this info in the supplementary method section.
- 3) in figure 3, western blots of markers of EMT should be included to complement the IFs. same is true for figure 3c.

Reviewer #3 (Remarks to the Author):

It is known that TGF- β signaling can induce epithelial-to-mesenchymal transition (EMT) and stemness in PDAC. In this study, Ottaviani and colleagues attempted to answer the question of whether microRNAs were regulated during the TGF- β response that in turn controls the TGF- β signaling network in PDAC. Using a gradient of cell lines displaying from epithelial-like to mesenchymal-like status, they found that the TGF- β /Smad2/3 axis induces transcription of MIR100HG, a long non-coding RNA and the host gene of miR-100, miR-125b and let-7a. Mechanistically, TGF- β induces LIN28B to inhibit Let-7a maturation, so that only miR-100 and miR-125b were upregulated upon TGF- β treatment. Functionally, both miR-100 and miR-125b cooperate to regulate similar pathways to promote stemness, EMT and tumorigenesis in PDAC. Overall the findings are interesting and potentially important, however, the authors should provide more direct and compelling evidence to support their conclusions. There are also some concerns that need to be clarified.

1. In Figure 1b-f, the authors use a gradient of cell lines from epithelial-like to mesenchymal-like status, aiming to find the miRNAs that are regulated by TGF- β . Although miR-100 and miR-125b were increased by TGF- β treatment in PANC-1 cells, there is no evidence showing that TGF- β is directly involved in regulating the differential expressions of miR-100 and miR-125b between epithelial-like BxPC-3 cells and mesenchymal-like S2-007 cells. To resolve this problem, the authors should determine miR-100 and miR-125b expressions in BxPC-3 cells untreated or treated with TGF- β , or in S2-007 cells untreated or treated with TGF- β /Smad signaling inhibitor.

2. In Figure 2a, it shows that SMAD2/3 interaction is localized close to the TSS of MIR100HG, the host gene of miR-100 and miR-125b, by analyzing RNA-seq and CHIP-seq results. However, the authors should provide direct evidence to draw the conclusion that SMAD2/3 binding to the host gene of miR-100 and miR-125b is regulated by TGF- β . For example, they should examine both miR-100 and miR-125b levels in PDAC cells expressing non-targeting control sh/siRNA or sh/siRNA targeting SMAD2/3 under TGF- β treatment and the interaction of SMAD2/3 with TSS is dynamically modulated.

3. It's well known that activation of LIN28 is responsible for the downregulation of let-7 miRNA family in many types of cancers. To draw a more compelling conclusion that LIN28 represses TGF- β -induced Let-7b, the authors should measure the expression of miR-100, miR-125b and Let-7b in LIN28B knockdown or control PDAC cells under TGF- β treatment. Another concern is whether LIN28A expression and function changes during this process.

4. In Figure 3d-f, the authors performed wound healing scratch assays to show that potential metastatic ability in PDAC cells was decreased with silencing of miR-100 or miR-125b. The next question is whether knockdown of miR-100 or miR-125b can reverse TGF- β -induced EMT and

metastasis.

5. It shows that overexpression of either miR-100 or miR-125b can significantly increase the number of tumor-spheres in Figure 4a. Since activation of TGF- β elevates both miR-100 and miR-125b expression levels, the authors need to explain why inhibition only one of the two miRNAs totally impairs the ability of TGF- β to increase PDAC tumour-spheres in Figure 4d.

6. In Figure 4g-j, the authors find that knockdown of miR-100 and miR-125b exhibited a strong reduction in tumor-initiating capacity. However, it would be important for the authors to determine whether silencing of miR-100 or miR-125b or both impairs the pro-tumorigenic ability of TGF- β in PDAC tumors.

Reviewer #1

In this study, Ottaviani et al characterize the effect of TGF- β signaling on specific microRNAs. While the results are provocative and point out a novel way in which TGF- β signaling modulates the phenotype of PDAC cells, the study as written could be improved with attention to the following:

We thank the referee for his/her kind remarks.

1. The authors correctly point out in their introduction that SMAD4 is a frequently mutated tumour suppressor gene in PDAC that acts in the TGF- β pathway. Thus, one could hypothesize that the downstream mediators of TGF- β signaling might differ between SMAD4 wildtype and mutant PDACs. It does not seem that the authors consider this possibility. Can they comment on possible differences in microRNA signaling between these two groups? Are the cell lines utilized in their experiments SMAD4 mutant or wildtype? Perhaps SMAD4 mutation is not important here since the authors suggest that the regulation is directly through SMAD2/3, but they should at least discuss this issue.

Considering the importance of SMAD4 in both the TGF- β response and pancreatic ductal adenocarcinoma (PDAC), we agree that evaluation of differences in TGF- β -mediated regulation of these two miRNAs in SMAD4 wild-type and mutant cells is a very important point and needs to be addressed. We apologise for this omission in the first version of our paper. We now show that the TGF- β -mediated induction of these two miRNAs seems to occur only in human PANC-1 and COLO357 cells, as well as in mouse CHX45 PDAC cells, which are all wild-type for SMAD4, but not in S2-007 and BxPC-3 cells which do not express SMAD4. We have now also added new experiments to address this point (please see the response to the first point raised by Reviewer #3), and have discussed these new results accordingly.

2. The authors use the terms EMT and CSC very loosely. In my opinion, they should be more precise in the phenotypes they are characterizing. For example, the authors claim that overexpression of their micro-RNAs induces EMT - as evidence they use morphology, expression and localization of 2 proteins, and motility in a wound healing assay. To this reviewer, it is acceptable to raise the possibility that these specific phenotypes are characteristic of EMT in the Discussion, but the results should not stretch the interpretation of these findings. Similarly, the authors claim that their microRNAs are important for "CSC formation" but then use a small number of very specific assays to

assess this possibility. While the results are interesting and have implications about the functional role of the microRNAs, I do not agree that they are specifically indicative of the role of cancer stem cells. Such speculation is better left in the Discussion.

We agree with the reviewer. We have now softened our interpretations, and report that our findings give an indication of the phenotypes observed, avoiding any over-interpretation of our results.

3. Since cancer stem cells are thought to represent a small proportion of cells in a primary tumour with a specific phenotype, it is unclear whether the reported cell line experiments can really address the role of the microRNAs in this phenotype. How can a cell line recapitulate the functional heterogeneity inherent in the cancer stem cell hypothesis?

This is true. CSC represent a very small proportion of cells within a cancer cell population, and a cell line cannot recapitulate the functional heterogeneity of a tumour. We have now re-phrased the results referring to tumourigenesis, instead of CSC formation.

4. The correlation of micro-RNA expression with OS is quite interesting. Can the authors also correlate with grade of differentiation of the primary tumours? Are those with high micro-RNA expression less differentiated/more mesenchymal in morphology on histological sections? This is important for the clinical relevance of the findings.

We thank the referee for this suggestion. We have now performed this analysis and discovered that only high miR-125b, and not high miR-100, was significantly associated with higher tumour grade of differentiation. We have added this new finding to the manuscript.

5. The authors suggest that these microRNAs might be therapeutic targets. Have any strategies been reported for targeting microRNAs *in vivo*? If so, these studies should be referenced. Otherwise, such a clinical application is a stretch, since targeting these is likely quite challenging.

Several strategies for miRNA targeting in cancer and other diseases have been recently reported. Evidences indicating the possibility to target miRNAs for therapeutic purposes are not limited to *in vivo* pre-clinical studies. Several clinical trials that use targeting of miRNAs for therapeutic are currently ongoing (reviewed in Christopher et al, 2016). We now accordingly reference some of these studies in the revised version of this manuscript.

Minor comments:

1. The following statement in the introduction is not accurate: "In PDAC, 95% of cases show mutational hyper-activation of KRAS and inactivation of the tumour suppressors TP53, P16/INK4A, and the TGF- β effector, SMAD4". 95% of PDACs have KRAS mutations, but the prevalence of tumour suppressor gene alterations is much lower - SMAD4 is altered in only ~50% of PDACs.

We thank the referee for this correct observation. We have now updated this information accordingly to the most recent genome sequencing studies that have revisited the PDAC mutational landscape (Waddel et al, Nature, 2015; Bailey et al, Nature, 2016).

2. I think a word has been omitted from this heading: "TGF- β increases miR-100 and miR-125b inducing MIR100HG transcription through SMAD2/3" Should it say "BY inducing"? As written it suggests that miR-100 and mir-125b induce MIR100HG, which is not supported by the data.

We are sorry for this grammatical error. This sentence has now been corrected. As the reviewer correctly pointed out "By inducing" is the correct way of phrasing this sentence.

Reviewer #2

In this manuscript, Ottaviani and colleagues propose a novel mechanism through which TGF-beta promotes EMT and cancer stem cell in pancreatic cancer.

According to this model, TGF- β eta transcriptionally activates a the miR-100~let7a2 polycistronic miRNA cluster, which encodes for miR-100, miR-125b, and let-7a2. However, TGF- β eta seem to also induce expression of LIN28 (an inhibitor of let-7 processing) and therefore while the levels of mature miR-100 and miR-125b increase, let-7a2 levels remain largely unchanged.

The selective upregulation of miR-100 and miR-125b, in turn leads to the direct and indirect modulation of a large set of genes, promoting EMT and stemness of pancreatic cancer cell lines.

The manuscript is clearly written, although the section on RIP-USE would benefit from some editing, and the topic is of substantial interest. The experiments are clearly described and the appropriate controls are included.

Thank you for these kind, and encouraging remarks.

We have now significantly improved the clarity of the RIP-USE methods section.

My major concern with this work is that the authors do not directly test the central claim of the paper: that induction of miR-100/125b by TGF- β is important for EMT and cancer cell 'stemness'. To be more precise: in fig. 1 they show that TGF- β eta treatment of PANC1 cells leads to a rather modest (less than 2 fold) increase in miR-100b and miR-125b. although it is certainly possible that even a modest increase in a miRNA can have substantial phenotypic consequences, this need to be experimentally demonstrated.

To support their model the authors performed two key experiments:

a) ectopic expression of miR-100 and miR-125b in BxPC-3, PANC-1 and CHX45 cells. Although these experiments seem to show a partial induction of EMT (Fig 2Am, B), miR-100 and miR-125b are likely induced to much higher levels compared to what the authors report in response to TGF- β eta treatment, and therefore they do not directly address the central hypothesis of the manuscript.

b) The second key experiment is using the miRZIP platform to inhibit miR-100 and miR-125b in pancreatic cancer cells expressing high levels of these miRNAs. Also in this case, the results show that inhibiting these miRNAs can impair cell migration (Fig 2D-F) and induce a more 'epithelial'

morphology (Fig 2C), but bear little relevance to the model proposed by the investigators.

c) more convincing are the experiments reported in fig. 4, in which miR-100 and miR-125b inhibitors are shown to impair the ability of TGF- β eta to increase the sphere formation efficiency of PANC-1 cells.

Although these experiments certainly link miR-100 and miR-125b to EMT and CSCs, they do not directly test the ability of TGF- β eta to induce EMT and CSCs in the via selective miR-100/miR125b induction. It must be pointed out that several papers have already linked miR-100 and miR-125b to EMT (for example PMID: 24586203, 24805183, 27383536) although a consensus on whether these miRNAs promote or antagonize EMT has not emerged yet.

In summary, the relevance and quality of this manuscript would be greatly improved if the authors included experiments directly testing the ability of TGF- β eta to promote EMT and CSCs via miR-100/miR-125b induction. More specifically, I would recommend performing the following relatively straightforward experiment:

Using CRISPR-Cas9, the investigators should delete the two Smad2/3 binding sites in the MIR100HG promoter region (Fig. 2a) in PANC-1 cells (or in another suitable model system). This should selectively impair the ability of TGF- β eta to induce miR-100/miR-125b. Alternatively, the investigators could should simply delete the two miRNA precursors.

They should then test the ability of TGF- β eta to induce partial EMT and promote CSC in these cells compared to control cells with the intact locus. This simple experiment should conclusively test the central hypothesis of the manuscript, complementing the other data presented, and overcoming the intrinsic limitations of overexpression and miRZIP studies.

We thank the reviewer for these insightful suggestions that have permitted us to greatly improve our manuscript.

As the reviewer suggested, we have now used CRISPR/CAS9 technology to create PANC-1 independent clones that do not express miR-100 or miR-125b (new fig. 4a), in order to test whether these clones have impaired TGF- β induced EMT and motility. This experiment has permitted us to test the central hypothesis of the manuscript, overcoming the intrinsic limitations of exogenous overexpression and miRZip technology. This was a very productive exercise, as by using these clones we now see that miR-125b represents the most important miRNA, amongst the miRNAs expressed

by MIR100HG, in modulating the tumourigenic response of TGF- β . This was true both in our *in vitro* and *in vivo* studies (see please also the response to both the penultimate and last points of the Reviewer #3). In addition, we have now demonstrated that silencing miR-100 and miR-125b reverses the ability of TGF- β to induce EMT and motility (new figs. 4b and c).

As the reviewer suggested, we also used CRISPR/CAS9 to remove the genomic region that encompasses the SMAD2/3 interacting regions around the MIR100HG transcription start site (TTS), that were predicted by MACS2 (<https://github.com/taoliu/MACS>) in both ChIP-seq replicates (Fig. 1 of the response to the reviewers, red arrows). This would theoretically impair the ability of TGF- β to induce the miRNAs. We were very excited to perform this remarkable experiment. This assay showed that removal of this genomic region significantly reduces MIR100HG, along with miR-100 and miR-125b expression, compared to controls (Fig. 2 of the response to the referees), indicating that it is important for miRNA transcription. Unexpectedly though, deletion of this region did not impede to TGF- β to induce both miRNAs (Fig. 3 of the response to the referees). This is probably due to additional and important SMAD2/3 interaction sites that may be located throughout the entire MIR100HG locus, which have not been predicted by MACS2 in one of the two replicates (false positives) (Fig. 1 of the response to the reviewers, green arrows). This is in line with the finding that pancreatic cancer cells from mice also have numerous SMAD2/3 binding sites throughout the MIR100HG transcript (Supplementary Fig. 2g), indicating a conserved, complex mode of SMAD2/3-mediated regulation of MIR100HG. In addition, we noticed that MIR100HG may have additional unannotated TTSs (Fig. 1 of the response to the reviewers, H3K27ac and H3K4me3 peaks), which adds to the complexity of the transcriptional regulation of this locus. Since the detailed study of MIR100HG transcriptional regulation is out of the scope of this study, and because the two miRNAs were still activated by TGF- β despite the created deletion, this experiment has not been included in the revised version of the manuscript.

Fig. 1 of response to the reviewers. Schematic of the experiment performed to delete the region containing SMAD2/3 interaction sites from the promoter of MIR100HG using CRISPR/CAS9.

Fig. 2 of response to the reviewers. RT-qPCR expression of MIR100HG, miR-100 and miR125b in CRISPR clones with deletion in the promoter region of MIR100HG

Fig. 3 of response to the reviewers. RT-qPCR expression of miR-100 and miR125b in CRISPR clones with deletion in the promoter region of MIR100HG after vehicle or TGF-β treatment.

Additional points:

1) A western blot showing LIN28 induction in response to TGF beta should be included

We have now included an immunoblot showing that, along with its mRNA, LIN28 protein is induced after 6h of TGF- β treatment (new fig. 2b).

2) the scatter plot shown in fig. 2 are a bit confusing. It seems that the majority of samples have low levels of LIN28. The criteria for defining low or high LIN28 levels, as well the total number of samples in each category and the correlation test used should be specified. I couldn't find this info in the supplementary method section.

We apologies for this oversight. We have now added the TCGA correlation analysis methods in the Supplementary Methods section. For this we used level 3 gene expression data from TCGA pancreatic datasets, which reports RSEM normalization data. The expression of LIN28B in the various samples ranged from 0 to 6.8 RSEMs, and we considered low LIN28B level samples to be those that express less than 0.5 RSEMs.

Indeed, the majority of the samples had LIN28B expression levels of less than 0.5 RSEMs. We used the Pearson correlation here and have added this information to the fig. legend.

3) in fig. 3, western blots of markers of EMT should be included to complement the IFs. same is true for fig. 3c.

We have now added immunoblots for markers of EMT to complement the IFs. Overall, we do not observe significant changes in E-cadherin levels in any of the cells when down-regulating / over-expressing miR-100 and miR-125b (new supplementary figs. 4a, b). This is in line with our general observation that miR-100 and miR125b do not induce EMT acting through the regulation of E-cadherin, but instead impair cell-cell junction interactions in an alternative way. Accordingly, our RIP-USE approach identified that miR-125b directly represses several claudins and other components of the tight and adherent junctions, which are crucial for cell-cell interactions. Down-regulation of these transcripts could induce EMT independently of E-cadherin levels. However, it remains less clear how miR-100 exactly performs this action. We have presented this new data in the revised version of the manuscript, as well as in the new Discussion.

Reviewer #3

It is known that TGF- β signaling can induce epithelial-to-mesenchymal transition (EMT) and stemness in PDAC. In this study, Ottaviani and colleagues attempted to answer the question of whether microRNAs were regulated during the TGF- β response that in turn controls the TGF- β signaling network in PDAC. Using a gradient of cell lines displaying from epithelial-like to mesenchymal-like status, they found that the TGF- β /Smad2/3 axis induces transcription of MIR100HG, a long non-coding RNA and the host gene of miR-100, miR-125b and let-7a. Mechanistically, TGF- β induces LIN28B to inhibit Let-7a maturation, so that only miR-100 and miR-125b were upregulated upon TGF- β treatment. Functionally, both miR-100 and miR-125b cooperate to regulate similar pathways to promote stemness, EMT and tumorigenesis in PDAC. Overall the findings are interesting and potentially important, however, the authors should provide more direct and compelling evidence to support their conclusions. There are also some concerns that need to be clarified.

1. In Fig. 1b-f, the authors use a gradient of cell lines from epithelial-like to mesenchymal-like status, aiming to find the miRNAs that are regulated by TGF- β . Although miR-100 and miR-125b were increased by TGF- β treatment in PANC-1 cells, there is no evidence showing that TGF- β is directly involved in regulating the differential expressions of miR-100 and miR-125b between epithelial-like BxPC-3 cells and mesenchymal-like S2-007 cells. To resolve this problem, the authors should determine miR-100 and miR-125b expressions in BxPC-3 cells untreated or treated with TGF- β , or in S2-007 cells untreated or treated with TGF- β /Smad signaling inhibitor.

We thank the reviewer for this interesting observation. Accordingly, we treated BxPC-3 and S2-007 with TGF- β , vehicle or TGF- β inhibitor, to evaluate whether miR-100 and miR-125b are induced by this signalling (new supplementary fig. 3g). This experiment was highly informative. Although S2-007 cells express high levels of miR-100 and miR-125b, at least 40-fold higher than their levels in BxPC-3, the miR-100 and miR-125b levels did not change with these treatments (new supplementary fig. 3g). We also observed that both these cells are SMAD4(-), whereas all the other PDAC cells in which we show a significant induction of miR-100 and miR-125b by TGF- β (i.e. PANC-1, COLO357 and CHX45) are SMAD4(+). We hypothesized that the presence of SMAD4 is crucial for the regulation of miR-100 and miR-125b by TGF- β . This is in line with a previous finding (David et al, Cell, 2016) which indicates that the regulation of EMT related genes by TGF- β requires SMAD4. In addition, RNAseq gene expression profiling showed a significant reduction in SMAD2 and SMAD3 for S2-007 versus BxPC-3, suggesting that it is unlikely that TGF- β signaling is involved in the up-regulation of miR-100 and miR-

125b expression in this mesenchymal-like cell line. This suggests that in absence of SMAD4, alternative oncogenic pathways, but not TGF- β , induces the expression of these miRNAs. We have added all this new data and discussion to the revised manuscript.

2. In Fig. 2a, it shows that SMAD2/3 interaction is localized close to the TSS of MIR100HG, the host gene of miR-100 and miR-125b, by analyzing RNA-seq and CHIP-seq results. However, the authors should provide direct evidence to draw the conclusion that SMAD2/3 binding to the host gene of miR-100 and miR-125b is regulated by TGF- β . For example, they should examine both miR-100 and miR-125b levels in PDAC cells expressing non-targeting control sh/siRNA or sh/siRNA targeting SMAD2/3 under TGF- β treatment and the interaction of SMAD2/3 with TSS is dynamically modulated.

The reviewer is completely correct. The binding of SMAD2/3 within the promoter of MIR100HG does not conclusively demonstrate that TGF- β is inducing the expression of this transcript or miR-100/125b through SMAD2/3. As the reviewer suggested, we have now silenced both SMAD2 and SMAD3 by using specific siRNAs (new supplementary fig. 2d) before TGF- β treatment. Silencing of SMAD2 or SMAD3 significantly reduced the levels of miR-100/125b compared with negative control, but more importantly it completely impaired the capacity of TGF- β to induce both miRNAs (new supplementary fig. 2f), conclusively proving that TGF- β controls the expression of these miRNAs through SMAD2/3.

3. It's well known that activation of LIN28 is responsible for the downregulation of let-7 miRNA family in many types of cancers. To draw a more compelling conclusion that LIN28 represses TGF- β -induced Let-7b, the authors should measure the expression of miR-100, miR-125b and Let-7b in LIN28B knockdown or control PDAC cells under TGF- β treatment. Another concern is whether LIN28A expression and function changes during this process.

To conclusively prove that LIN28B is induced by TGF- β in order to repress let-7a, we created PANC-1 clones that were KO for LIN28B using CRISPR/CAS9 technology (new supplementary fig. 3e). Strikingly, TGF- β was able to significantly induce let-7a in the LIN28B KO clones, but not in the control cells, thus proving our hypotheses (new supplementary fig. 3e). We also show that LIN28A was never expressed or induced in any of these cells (treated or untreated with TGF- β), but it is expressed in IMR90 which is a positive control cell line expressing LIN28A (new supplementary fig.

3e).

4. In Fig. 3d-f, the authors performed wound healing scratch assays to show that potential metastatic ability in PDAC cells was decreased with silencing of miR-100 or miR-125b. The next question is whether knockdown of miR-100 or miR-125b can reverse TGF- β -induced EMT and metastasis.

We have now evaluated cellular shape to test EMT as well as wound healing (motility) assays after treating the cells with TGF- β prior to transient inhibition of miR-100, miR-125b or negative controls (using anti-miRNA molecules). As shown in the new figs. 4b and c, the inhibition of both miRNAs reversed TGF- β -induced EMT and motility.

5. It shows that overexpression of either miR-100 or miR-125b can significantly increase the number of tumour-spheres in Fig. 4a. Since activation of TGF- β elevates both miR-100 and miR-125b expression levels, the authors need to explain why inhibition only one of the two miRNAs totally impairs the ability of TGF- β to increase PDAC tumour-spheres in Fig. 4d.

An explanation of this could be drawn from our miRNA-target discovery pipeline, RIP-USE, which revealed that, these two miRNAs, target several common transcripts, as well as genes that in general are involved in controlling the same pathways. This indicates that if TGF- β regulates crucial targets involved stemness via these two miRNAs (e.g. two different transcripts coding for interacting proteins, or even the same crucial transcript), then inhibition of either miRNA would impede TGF- β to regulate these phenotypes. This same hypothesis could be valid for motility and EMT. We must point out that we have now repeated the sphere-forming assays using newly generated cells KO for the expression of these miRNAs by CRISPR/CAS9. Consequently, we clearly observe that KO clones for miR-125b completely reverse the TGF- β response, although KO clones for miR-100 were still able to increase their sphere-tumour capacity mediated by TGF- β . These new results were obviously unexpected. Since we could not reproduce the effect of miR-100 between different experiments, but we clearly observe a constant and reproducible reversion of the TGF- β mediated phenotype by miR-125b, we now hypothesize that miR-125b represents the most important TGF- β effector amongst the miRNAs expressed by MIR100HG. In fact, different strengths in the effect of miRNAs belonging to the same cluster in mediating particular phenotypes has been previously demonstrated. The Ventura lab has previously shown that that amongst the 6 miRNAs that are encoded by miR-17-92 transcript,

miR-19a and miR-19b are the only ones absolutely required, and are sufficient enough to recapitulate the oncogenic properties of the entire cluster (Mu et al, Genes and dev, 2019).

6. In Fig. 4g-j, the authors find that knockdown of miR-100 and miR-125b exhibited a strong reduction in tumour-initiating capacity. However, it would be important for the authors to determine whether silencing of miR-100 or miR-125b or both impairs the pro-tumourigenic ability of TGF- β in PDAC tumours.

We agree with the reviewer that evaluating whether silencing of miR-100 or miR-125b or their combination, impairs the pro-tumourigenic ability of TGF- β in PDAC tumours is an important point. To answer this question, we have now performed a new *in vivo* tumourigenic assay using PANC-1 controls or stably over-expressing TGF- β , with or without the inhibition of miR-100 and miR-125b. As expected, tumours formed from cells over-expressing TGF- β developed with higher frequency compared to cells over-expressing empty vectors (new supplementary fig. 5e). Interestingly, cells with reduced miR-125b activity completely reverted the TGF- β -mediated tumourigenic effect, but this was not true for cells with reduced miR-100 activity. This again indicated that miR-125b represents the main player in the TGF- β mediated tumourigenic effect in PDAC.

Thank you for your time and effort in reviewing our work, thanks to your suggestions and comments we trust that you now find the revised manuscript suitable for publication in *Nature Communications*.

Reviewers' comments:

Reviewer #1 (Remarks to the Author):

The authors have effectively addressed all of my concerns in the revised manuscript.

Reviewer #2 (Remarks to the Author):

I have read the authors' response to the reviewers comments as well as the revised manuscript. In my initial review, my major concern was that the authors had not directly tested the central claim of the manuscript: that transcriptional induction of miR100 and miR-125 is essential for the ability to TGF-beta to promote stemness and EMT in pancreatic adenocarcinoma cells. In particular I was concerned that the increase in miR-100 and miR-125 observed in response to TGF-beta treatment was rather modest (less than 2 fold) and that the authors did not provide sufficient evidence that such a modest increase was necessary for TGF-beta to induce EMT and increase tumorigenicity. To address this concern I recommended the authors to use CRISPR-Cas9 to genetically inactivate miR-100 and miR-125 or, even better, to mutate the proposed Smad2/3 binding sites in the miR100HG promoter region in PANC-1 cells, and then examine the consequence of TGF-beta treatment on EMT and 'stemness'.

I commend the authors for generating the targeted mutants, but I am confused by the results shown and by the lack of details presented in the revised manuscript, especially given that these results should provide key evidence in support of their model.

More specifically, the only results related to genetic inactivation of miR-100/miR-125 are shown in a single panel in figure 4 (Figure 4a). In the figure legend the authors indicate that they have generated three independent clones of miR-100 or miR-125 mutant cells and they show that sphere formation efficiency in response to TGF-beta treatment is reduced in both mutants, but more strongly in miR-125 mutant cells.

The issues I have are the following:

1) Scant details are provided on how the mutants were generated and, more problematic in my view, no validation of the targeted clones is provided showing bi-allelic loss of the corresponding miRNAs. At a minimum, a supplementary figure showing genotyping (sequencing) of the clones and RT-PCR of the mutated miRNAs should be included. These are key reagents that are central to the proposed model.

As an aside, the section on the generation of these mutant alleles in the revised method section is confusing (page 3 of supplementary methods file). A long list of sequences against a variety of genes has been added under the heading: "Oligonucleotides used for sgRNAs cloned in CRISPR/CAS9". To me, they all look like PCR primers and not sgRNAs sequence, and in fact they are identical to the oligos for PCR presented in the next section under the heading "RNA isolation and RT-qPCR assays".

2) It is unclear why only the results of sphere formation assays are reported with these cells. The experiments reported in figure 4b and 4c, at a minimum, should also be repeated using these mutant clones. These are simple, fast, and relatively inexpensive experiments and it makes little sense to use non-specific miRNA inhibitors when highly specific targeted alleles are available to the investigators.

In addition, the authors over-interpret the result of this sphere formation experiment by claiming that : "Remarkably, TGF-beta mediated tumorigenesis was reverted in vitro in KO PANC-1 clones lacking pro-tumorigenic miR-125b...". The experiment only shows that increased sphere formation is inhibited, not that 'tumorigenicity' is blocked.

3) I am also puzzled by the author's interpretation of the second experiment. Here they used CRISPR to delete the entire region of the miR-100HG promoter containing the two Smad2/3

binding sites that the authors claim in the manuscript are responsible for the modest upregulation of miR-100/miR-125 in response to TGF-beta (incidentally, my suggestion was to specifically delete only the sites, not the entire region including the TSS, as the authors did). The results indicate of these experiments, however, show that deletion of this region does NOT impair the ability of TGF-beta to upregulate miR-100/125 (reviewer figure 3). This is a key result that is unfortunately inconsistent with the model the authors propose in the paper.

The authors comment this result in the rebuttal letter saying: "This is probably due to additional and important SMAD2/3 interaction sites that may be located throughout the entire MIR100HG locus, which have not been predicted by MACS2 in one of the two replicates (false positives) (Fig. 1 of the response to the reviewers, green arrows)."

While this is certainly possible, it is equally possible that the upregulation of miR-100/125 is NOT due to Smad2/3 binding to the Mir100-HG promoter, but is an indirect effect (some other transcription factor induced by TGF-beta) or even post-transcriptional.

Surprisingly, the authors further comment saying that "Since the detailed study of MIR100HG transcriptional regulation is out of the scope of this study, and because the two miRNAs were still activated by TGF-β despite the created deletion, this experiment has not been included in the revised version of the manuscript."

I find the decision of omitting from the revised manuscript experiments inconsistent with the proposed model perplexing, especially because in the revised manuscript the authors continue to discuss the two SMAD2/3 binding sites and state in the abstract that "TGF-beta transcriptionally induces MIR100HG, containing miR-100, miR-125b and let-7 in its intron, via SMAD2/3" and in the main text that "In aggregate, these findings suggest that TGF-beta activates SMAD2/3, that in turn directly regulates the transcription of a gene network which includes MIR100HG along with MIR-100 and miR-125b..." and figure 8 continues to depict a model in which SMAD2/3 directly activate MIR100HG.

4) I thank the authors for providing additional information on the RIP-USE method. While the method is certainly interesting, it is unclear to this reviewer whether the results obtained are superior to what one would obtain by simply combining RNAseq, Sylamer, and Targetscan prediction. In other words, it is unclear whether the use of Ago2-RIP allows a more accurate identification of direct targets. I am asking this because the cdf plots shown in Figure 6 are not particularly striking (note that the x scale is compressed between -0.4 and +0.4 log2FC to facilitate the separation of the curves). The easiest way to address this point is to compare the ecdf plots of genes identified using the RIP-USE procedure and the ecdf plots of genes selected based on the presence of predicted miR-100/125 sites according to targetscan. Clarifying this point would be important to demonstrate that the RIP-USE method offers significant advantage over more conventional approaches.

Additional minor point: Line 417 should read : "transcripts containing 8mer, 7mer..." not "lacking".

Reviewer #3 (Remarks to the Author):

In this revised manuscript, the authors provided a number of comprehensive results and interpretations to address concerns raised in the review. The additional experiments further confirmed that TGFβ controls miR100 and miR125b expression by SMAD2/3. They also found that TGFβ induces LIN28B to repress let7. Moreover, the new in vivo tumourigenic test showed that miR125b represents the main player in the TGF-β mediated tumourigenic effect in PDAC.

Overall, this revised version is significantly improved and the authors have satisfactorily addressed all of my concerns.

Response to the reviewers

Reviewer #1 (Remarks to the Author):

The authors have effectively addressed all of my concerns in the revised manuscript.

We thank the referee for the time that he/she spent in revising our manuscript.

Reviewer #2 (Remarks to the Author):

I have read the authors' response to the reviewers comments as well as the revised manuscript. In my initial review, my major concern was that the authors had not directly tested the central claim of the manuscript: that transcriptional induction of miR-100 and miR-125 is essential for the ability to TGF-beta to promote stemness and EMT in pancreatic adenocarcinoma cells. In particular I was concerned that the increase in miR-100 and miR-125 observed in response to TGF-beta treatment was rather modest (less than 2 fold) and that the authors did not provide sufficient evidence that such a modest increase was necessary for TGF-beta to induce EMT and increase tumorigenicity. To address this concern I recommended the authors to use CRISPR-Cas9 to genetically inactivate miR-100 and miR-125 or, even better, to mutate the proposed Smad2/3 binding sites in the miR100HG promoter region in PANC-1 cells, and then examine the consequence of TGF-beta treatment on EMT and 'stemness'.

In response to the referees' concerns, we have now further demonstrated that the transcriptional induction of miR-100 and miR-125b exerted by TGF- β is important for the ability of TGF- β to promote stemness and EMT in pancreatic ductal adenocarcinoma (PDAC) cells. As suggested by the referee, we have used our CRISPR miR-100 and miR-125b KO clones to show that miR-100 or miR-125b KO greatly impairs the ability of TGF- β to mediate tumour-sphere formation, and also EMT and motility, which is now included in this new revision (**please see response to comment 2**). Regarding the magnitude of miR-100 and miR-125b induction by TGF- β , we have examined several independent RT-qPCR experiments, as well as our nCounter miRNA expression profiling data, and observed that TGF- β increases the level of miR-100 and miR-125b up to an average of two-fold (not less) and this is consistent in all the TGF- β responsive PDAC cell lines tested. Importantly, by using different approaches of miRNA inhibition (anti-miRNAs, CRISPR KOs and miR-Zips), we have now

conclusively demonstrated that this two-fold induction of miR-100 and miR-125b by TGF- β is very important for the TGF- β response, both *in vitro* and *in vivo*. Therefore, we believe that the two-fold induction of miRNA expression can have important biological consequences, and this is also in line with other studies. For example, we and others have demonstrated that an induction of only 1.5-1.7 of miR-17 by Estrogen Receptor alpha (ER-alpha) in breast cancer is important for ER-alpha transcriptional activity and its biological response (Bhat-Nakshatri et al, Nucleic Acids Research, 2009; Castellano et al, PNAS, 2009). Moreover, Xiao et al (Cell, 2007) elegantly demonstrated that just 30-35% reduction in MYB protein in mice, due to miR-150 regulation, is sufficient to cause a dramatic effect on the numbers of mature B cells in the spleens of mice models.

I commend the authors for generating the targeted mutants, but I am confused by the results shown and by the lack of details presented in the revised manuscript, especially given that these results should provide key evidence in support of their model.

More specifically, the only results related to genetic inactivation of miR-100/miR-125 are shown in a single panel in figure 4 (Figure 4a). In the figure legend the authors indicate that they have generated three independent clones of miR-100 or miR-125 mutant cells and they show that sphere formation efficiency in response to TGF-beta treatment is reduced in both mutants, but more strongly in miR-125 mutant cells.

The issues I have are the following:

1) Scant details are provided on how the mutants were generated and, more problematic in my view, no validation of the targeted clones is provided showing bi-allelic loss of the corresponding miRNAs. At a minimum, a supplementary figure showing genotyping (sequencing) of the clones and RT-PCR of the mutated miRNAs should be included. These are key reagents that are central to the proposed model.

As an aside, the section on the generation of these mutant alleles in the revised method section is confusing (page 3 of supplementary methods file). A long list of sequences against a variety of genes has been added under the heading: "Oligonucleotides used for sgRNAs cloned in CRISPR/CAS9". To me, they all look like PCR primers and not sgRNAs sequence, and in fact they are identical to the oligos for PCR presented in the next section under the heading "RNA isolation and RT-qPCR assays".

We apologies for not providing important details about the miR-100 and miR-125b CRISPR-KO clones in the previous version of our manuscript. This information is now provided in full in **figure 4a and supplementary figure 7 a-d**. As shown in these figures, we have used two flanking sgRNAs for each

pre-miRNA to disrupt miRNA expression, in order to delete part or the entire miRNA locus. As the referee and the editors can observe from **supplementary figure 7a**, we were able to completely remove the pre-miR-100 genomic region from all the alleles using this approach. Regarding miR-125b, the two flanking sgRNAs used were unable to delete the locus in between (**supplementary figure 7b**). Nevertheless, since any disruption in this region should prevent the correct folding of the precursor, and therefore the ability of Drosha or Dicer to recognize it as a substrate, this strategy successfully KO miR-125b in all the alleles of PANC-1 cells. The success of the KO for both miRNAs has been further demonstrated by RT-qPCR of miR-100 and miR-125b in these clones (**supplementary figure 7 c-d**). Indeed, it is sufficient to use a single disruption within the region that forms the stem-loop, using a single sgRNA, to completely KO miRNA expression, as previously described (Kim et al Nat Struct Mol Biol, 2013). This is due to inefficient NHEJ repair after double-strand DNA break that produces a different sequence in the precursor region which then becomes unable to fold correctly. As expected, using a pair of sgRNAs we were able to increase the ability to completely KO miRNA expression and in all the alleles.

We apologize again for the incorrect and scant information provided in the Methods section. This issue has now been addressed, and we have also included the correct sgRNA sequences in **page 4 and 5 of the supplementary methods**.

2) It is unclear why only the results of sphere formation assays are reported with these cells. The experiments reported in figure 4b and 4c, at a minimum, should also be repeated using these mutant clones. These are simple, fast, and relatively inexpensive experiments and it makes little sense to use non-specific miRNA inhibitors when highly specific targeted alleles are available to the investigators.

In addition, the authors over-interpret the result of this sphere formation experiment by claiming that : “Remarkably, TGF-beta mediated tumorigenesis was reverted in vitro in KO PANC-1 clones lacking pro-tumorigenic miR-125b...”. The experiment only shows that increased sphere formation is inhibited, not that ‘tumorigenicity’ is blocked.

As the referee suggested, we have now repeated experiments previously reported in **figure 4b and 4c** with 3 independent miR-100 and miR-125b CRISPR KO clones. Using this strategy, we further demonstrate that TGF- β mediated EMT and motility is greatly impaired when miR-100 or miR-125b are inhibited (**new figure 4c-e**). We agree with the referee that the experiment with KO PANC-1 clones cannot show an effect on tumourigenesis, and that this would be an over-interpretation. We

have now changed the text accordingly indicating that “TGF- β -mediated increase in tumour-sphere formation was reverted *in vitro* in independent PANC-1 KO clones for miR-125b”.

3) I am also puzzled by the author’s interpretation of the second experiment. Here they used CRISPR to delete the entire region of the miR-100HG promoter containing the two Smad2/3 binding sites that the authors claim in the manuscript are responsible for the modest upregulation of miR-100/miR-125 in response to TGF-beta (incidentally, my suggestion was to specifically delete only the sites, not the entire region including the TSS, as the authors did). The results indicate of these experiments, however, show that deletion of this region does NOT impair the ability of TGF-beta to upregulate miR-100/125 (reviewer figure 3). This is a key result that is unfortunately inconsistent with the model the authors propose in the paper.

The authors comment this result in the rebuttal letter saying: “This is probably due to additional and important SMAD2/3 interaction sites that may be located throughout the entire MIR100HG locus, which have not been predicted by MACS2 in one of the two replicates (false positives) (Fig. 1 of the response to the reviewers, green arrows).”

While this is certainly possible, it is equally possible that the upregulation of miR-100/125 is NOT due to Smad2/3 binding to the Mir100-HG promoter, but is an indirect effect (some other transcription factor induced by TGF-beta) or even post-transcriptional.

Surprisingly, the authors further comment saying that “Since the detailed study of MIR100HG transcriptional regulation is out of the scope of this study, and because the two miRNAs were still activated by TGF- β despite the created deletion, this experiment has not been included in the revised version of the manuscript.”

I find the decision of omitting from the revised manuscript experiments inconsistent with the proposed model perplexing, especially because in the revised manuscript the authors continue to discuss the two SMAD2/3 binding sites and state in the abstract that “TGF-beta transcriptionally induces MIR100HG, containing miR-100, miR-125b and let-7 in its intron, via SMAD2/3” and in the main text that “In aggregate, these findings suggest that TGF-beta activates SMAD2/3, that in turn directly regulates the transcription of a gene network which includes MIR100HG along with MIR-100 and miR-125b...” and figure 8 continues to depict a model in which SMAD2/3 directly activate MIR100HG.

We apologies to the referee. He/she is completely correct. We have erroneously left unchanged our previous observation in the text of the manuscript, that SMAD2/3 certainly use sites located within the MIR100HG promoter to regulate MIR100HG transcription. Nevertheless, as we indicated in the

previous revision, SMAD2/3 bind to several genomic sites along the MIR100HG locus in both humans and mice (**supplementary figure 2d and supplementary figure 3b**). Thanks to the referee' s observation, we have now decided to include these experiments and data in the main manuscript, as we believe it could be very informative for the readers. Importantly, we do not find the results from our CRISPR deletion of the SMAD2/3 predicted sites on MIR100HG to be inconsistent with our model. We still suggest that the regulation of MIR100HG transcription occurs via SMAD2/3, because of several observations. 1) SMAD2/3 silencing by siRNAs completely impedes TGF- β to regulate miR-100 and miR-125b levels (**see supplementary figure 2f**). This effect cannot be due to post-transcriptional regulation of miRNA processing because RNA-seq shows that the entire MIR100HG transcript is regulated by TGF- β (**Figure 2a and Supplementary Table 2**). 2) We now show that the only TGF- β regulated transcript situated in the topological associated domain (TAD), where MIR100HG and SMAD2/3 chromatin binding sites are located (analysis of publicly available Hi-C data from PANC-1; **see supplementary figure 3a**), is MIR100HG, strongly suggesting that SMAD2/3 bind in this region to specifically regulate its transcription. 3) In addition, we show that miR-100 and miR-125b levels start increasing very early after the TGF- β treatment, indicating direct regulation by the TGF- β activated factors (i.e. SMAD2/3) (**see figure 2b**). In aggregate, we observe that MIR100HG has a complex transcriptional regulation, which is more complex than previously thought, with several transcriptional initiation sites. We now include these observations in the new version of the manuscript. We show that when we remove the first transcriptional start site, containing the strongest SMAD2/3 binding regions, although this reduces miR-100 and miR-125b levels, it does not impair TGF- β regulation. We suggest that this may be due to new or stronger regulation through the other SMAD2/3 binding sites along this locus. This could encourage future studies to investigate the regulation of MIR100HG or other transcripts, which would now be possible thanks to the advent of genome editing approaches. Accordingly, we removed SMAD2/3 interaction with MIR100HG promoter from our proposed model in **figure 8**, and instead we indicate that SMAD2/3 can interact with this locus in multiple regions, and have discussed that the way TGF- β /SMAD2/3 signaling regulates MIR100HG is still unclear and need further investigation.

4) I thank the authors for providing additional information on the RIP-USE method. While the method is certainly interesting, it is unclear to this reviewer whether the results obtained are superior to what one would obtain by simply combining RNAseq, Sylamer, and Targetscan prediction. In other words, it is unclear whether the use of Ago2-RIP allows a more accurate identification of direct targets. I am asking this because the cdf plots shown in Figure 6 are not particularly striking (note that the x scale is compressed between -0.4 and +0.4 log₂FC to facilitate

the separation of the curves). The easiest way to address this point is to compare the ecdf plots of genes identified using the RIP-USE procedure and the ecdf plots of genes selected based on the presence of predicted miR-100/125 sites according to targetscan. Clarifying this point would be important to demonstrate that the RIP-USE method offers significant advantage over more conventional approaches.

We thank the referee for this comment. We have now provided additional information to further clarify the RIP-USE method, and to explain the advantages of this approach over more conventional ones. By combining AGO2-RIP with Sylamer or cWords and RNA-seq after overexpression of the miRNA of interest, one can obtain accurate identification of the direct targets and important information about the type of regulation. In particular, AGO2-RIP experimentally identifies the transcripts that are directly bound by the miRNA of interest that could be canonical or non-canonical. By only using RNA-seq, Sylamer or TargetScan, one could miss direct targets acting through 'frequent' non-canonical seeds. Importantly, we found that the use of Sylamer or cWords in RIP-seq data was superior to unbiased detection of miRNA interaction sites, compared to the RNA-seq experiment. In fact the use of Sylamer or cWords after RNA-seq did not show a clear significant signal for canonical miRNA interaction sites, indicating that by only using RNA-seq, we would have been unable to find the non-canonical sites of miRNA interaction with Sylamer (which in addition cannot be predicted by online tools such as TargetScan). Finally, RIP-USE uniquely allows the user to establish whether the identified enriched words (canonical or non-canonical) are actually capable of down-regulating transcripts. Indeed, CLIP and CLASH experiments have in fact demonstrated that selected miRNAs could interact with transcripts through 'seedless' regions, such as their 3'parts, more frequently than their 5'parts (see miR-92 in Helwak et al, Cell, 2013), but contrarily to our RIP-USE, CLIP and CLASH methods cannot indicate whether non-canonical interactions are functional for target repression, because these methods are not combined with RNA-seq. In our study, after AGO2-RIP on miR-100 or miR-125b overexpressing cells, Sylamer and cWords predicted enrichment of only canonical seeds for miR-100 and miR-125b. As proof-of-principle, we use ecdf plots to further confirm that the enriched words are indeed able to down-regulate transcripts (in this it was obvious as there are canonical seeds). This means that those ecdf plots shown in **figure 6 b,c** were constructed by using all the expressed transcripts that contain canonical words (not a selection identified by RIP-USE), in comparison to transcripts that do not contain such words. Importantly, using RIP-USE we could consider the overlap of regulated transcripts between RIP-seq and RNA-seq (**figure 7 a,b**), and therefore transcripts directly regulated by the two miRNAs. Again, this would not have been possible by using RNA-seq alone. Importantly, by using RIP-USE, and not by restricting

ourselves to only RNA-seq plus Sylamer, we found that miR-100 may directly regulate transcripts through a still unidentified way in addition to the canonical method, and this was in accord with CLASH experiments (**supplementary figure 11b**). Compressing the scale to see a clearer difference in ecdf plots when miRNA regulation is analyzed from RNA-seq data is a common practice for the evaluation of miRNA repression (see for example figure 4c in Kim et al, Nat Struct Mol Biol, 2013). Low signal is probably due to a general mild regulation of miRNA repression (Baek et al, Nature, 2008; Selbach et al, Nature, 2008).

Additional minor point: Line 417 should read :“transcripts containing 8mer, 7mer...” not “lacking”.

This has now been corrected.

Reviewer #3 (Remarks to the Author):

In this revised manuscript, the authors provided a number of comprehensive results and interpretations to address concerns raised in the review. The additional experiments further confirmed that TGF β controls miR100 and miR125b expression by SMAD2/3. They also found that TGF β induces LIN28B to repress let7. Moreover, the new in vivo tumorigenic test showed that miR125b represents the main player in the TGF- β mediated tumorigenic effect in PDAC.

Overall, this revised version is significantly improved and the authors have satisfactorily addressed all of my concerns.

We thank the referee for these comments and for the time that he/she spent in revising our manuscript.

REVIEWERS' COMMENTS:

Reviewer #2 (Remarks to the Author):

The authors have addressed the majority of my concerns in the revised manuscript.